# AutoGFM: Automated Graph Foundation Model with Adaptive Architecture Customization

Haibo Chen [1]  Xin Wang [1]  Zeyang Zhang [1]  Haoyang Li [1]  Ling Feng [1]  Wenwu Zhu [1]

## Abstract

Graph foundation models (GFMs) aim to share graph knowledge across diverse domains and tasks to boost graph machine learning. However, existing GFMs rely on hand-designed and fixed graph neural network (GNN) architectures, failing to utilize optimal architectures *w.r.t.* specific domains and tasks, inevitably leading to suboptimal performance in diverse graph domains and tasks. In this paper, we explore graph neural architecture search (GNAS) for GFMs for the first time, which suffers from the problem of *architecture inconsistency*, i.e., the optimal architectures for different tasks and domains vary. We tackle this problem by discovering an invariant graph-architecture relationship across domains and tasks, which imposes three challenges: i) how to capture invariant and variant patterns; ii) how to customize architectures to adapt to diverse domains and tasks; iii) how to mitigate the data domination phenomenon during the architecture search process. To address these challenges, we propose Automated Graph Foundation Model with Adaptive Architecture Customization (**AutoGFM**), providing a theoretical analysis to demonstrate the limitations of existing GNAS. Specifically, we first propose a disentangled contrastive graph encoder to learn invariant and variant patterns. Then, we design an invariant-guided architecture customization strategy to customize architectures for data from diverse domains and tasks. Finally, we propose a curriculum architecture customization mechanism to mitigate the phenomenon of particular data dominating the search process. Extensive experiments demonstrate that **AutoGFM** outperforms baselines, achieving state-of-the-art performance.

[1]Department of Computer Science and Technology, BNRIST, Tsinghua University, Beijing, China. Correspondence to: Xin Wang <xin_wang@tsinghua.edu.cn>, Wenwu Zhu <wwzhu@tsinghua.edu.cn>.

*Proceedings of the $42^{nd}$ International Conference on Machine Learning*, Vancouver, Canada. PMLR 267, 2025. Copyright 2025 by the author(s).

## 1. Introduction

Graph foundation models (GFMs) (Liu et al., 2023b; Xu et al., 2024; Kong et al., 2024) aim to share graph knowledge across diverse graph domains and tasks. GNN-based GFMs (Liu et al., 2023a; Wang et al., 2024b) represent a promising direction, as they enable the transfer of shared knowledge across various domains and tasks, allowing a single graph neural network (GNN) to handle node-level, edge-level, and graph-level tasks across various domains. Specifically, GNN-based GFMs leverage large language models (LLMs) as enhancers, transform the textual features of graphs into unified representations, and unify graph-related tasks through subgraph classification for GNNs.

However, data from different tasks and domains may require different graph neural architectures. For instance, the vanilla GCN (Kipf & Welling, 2017) outperforms GraphSAGE (Hamilton et al., 2017) in the citation network OGBN-arxiv, while failing to demonstrate satisfactory performance in OGBN-proteins (Hu et al., 2020). Since existing GNN-based GFMs rely on hand-designed and fixed GNN architectures, they inevitably fail to adapt to the specific architecture requirements for diverse domains and tasks.

In this paper, we explore the problem of graph neural architecture search (GNAS) for GNN-based graph foundation models, which suffers from the problem of *architecture inconsistency*, i.e., the optimal architecture for different tasks and domains varies. We further leverage a representative group of differentiable graph neural architecture search methods (Liu et al., 2018) as an example and provide theoretical analysis demonstrating their inability to effectively search for graph neural architectures for GFMs under *architecture inconsistency*, resulting in suboptimal architectures. We tackle this problem by discovering an invariant graph-architecture relationship across domains and tasks, which imposes three challenges: i) how to capture invariant and variant patterns, which are entangled in graph data; ii) how to customize graph neural architectures based on the discovered patterns to adapt to data with diverse domains and tasks; iii) how to mitigate the phenomenon of data domination during the architecture search process.

To address these challenges, we propose a novel Automated

Graph Foundation Model with Adaptive Architecture Customization (**AutoGFM**), which customizes graph neural architectures for graph data across diverse tasks and domains. The core idea is to train an architecture mapping function $\pi$, which maps $\mathcal{G}$ (graph data) $\rightarrow \mathcal{A}$ (architecture), enabling the customization of architectures for each dataset to address *architecture inconsistency*, while simultaneously facilitating mutual knowledge sharing across diverse domains and tasks within a weight-sharing super-network. Specifically, we first propose a disentangled contrastive graph encoder to learn invariant and variant patterns from graph data. To achieve this, we design a subgraph-level discriminative contrastive learning that captures the invariant and variant patterns from diverse graph data. Second, we propose an invariant-guided architecture customization to tailor graph neural architectures for diverse data. We encourage invariant patterns to retain their ability to customize architectures despite the interference from variant patterns, aiming to eliminate the spurious effects brought by variant patterns. Finally, we propose a curriculum architecture customization mechanism to mitigate the phenomenon of some particular data dominating the search process. We design a curriculum constraint to promote the diversity of customized architectures across different datasets. Extensive experiments demonstrate that our **AutoGFM** model outperforms existing baselines, achieving state-of-the-art performance. The contributions of this paper are summarized as follows:

- We propose to explore the problem of graph neural architecture search for GNN-based graph foundation model, to the best of our knowledge, for the first time.

- We propose Automated Graph Foundation Model with Adaptive Architecture Customization (**AutoGFM**), analyzing the problem of *architecture inconsistency* for GFM and providing a theoretical analysis to demonstrate the limitations of existing mainstream differentiable GNAS methods under such conditions.

- We propose three novel modules to tackle the problem of *architecture inconsistency*, i) disentangled contrastive graph encoder, ii) invariant-guided architecture customization, and iii) curriculum architecture customization mechanism.

- We conduct extensive experiments on eight datasets to demonstrate the superiority of our method over state-of-the-art baselines.

## 2. Problem Formulation

In this section, we introduce the fundamental concepts and notations used in this paper, including graph data definition, node of interest (NOI) graph, graph neural architecture search, and GNAS for GNN-based GFMs.

### 2.1. Graph Data Definition

**Text-attributed Graphs (TAGs)**  A text-attributed graph (TAG) is a graph where each node and edge is associated with a text sentence (Liu et al., 2023a). We denote a TAG is denoted as $\mathcal{G} = (\mathcal{V}, \mathcal{E}, \mathcal{R})$, where $\mathcal{V} = v_1, \ldots, v_{|\mathcal{V}|}$ represents the set of nodes, $\mathcal{E} = e_1, \ldots, e_{|\mathcal{E}|}$ represents the set of edges, and $\mathcal{R} = r_1, \ldots, r_{|\mathcal{R}|}$ represents the set of relations.

**Node of Interest (NOI) Subgraph**  Given a graph $\mathcal{G} = (\mathcal{V}, \mathcal{E}, \mathcal{R})$. Following the previous work (Liu et al., 2023a; Wang et al., 2024b), we define the subgraph to unify graph tasks as node of interest subgraph (NOI-graph). An NOI-graph $G_h$ is defined as the subgraph around the NOI. Denote $S_h(v) = \{\mathcal{V}_v^h, \mathcal{E}_v^h, \mathcal{R}_v^h\}$ as the $h$-hop ego-subgraph around $v$, consisting of $h$-hop neighbor nodes of $v$ and all interconnecting edges. For node-level tasks on a node $v$, the NOI is the node itself, such that $\mathcal{T} = \{v\}$ and $G_h(\mathcal{T}) = S_h(v)$. For link-level tasks involving a node pair $(v_i, v_j)$, we define $\mathcal{T} = \{v_i, v_j\}$, and the NOI-graph is $G_h(\{v_i, v_j\}) = S_h(v_i) \cup S_h(v_j)$. For graph-level tasks, the NOI includes all nodes in the graph, making the NOI-graph $G_h(\mathcal{V}) = (\mathcal{V}, \mathcal{E}, \mathcal{R})$. We define an NOI-graph $G_h(\mathcal{T})$ as:

$$G_h(\mathcal{T}) = \cup_{v \in \mathcal{T}} S_h(v)$$
$$= \big( \cup_{v \in \mathcal{T}} \mathcal{V}_v^h, \cup_{v \in \mathcal{T}} \mathcal{E}_v^h, \cup_{v \in \mathcal{T}} \mathcal{R}_v^h \big). \quad (1)$$

### 2.2. Graph Neural Architecture Search

Given a data $\mathcal{D} = (\mathcal{G}, \mathcal{Y})$ for a graph neural architecture search (GNAS), we aim to search for a function $F_{\alpha, w} : \mathcal{G} \rightarrow \mathcal{Y}$, with architecture parameters $\alpha \in \mathcal{A}$ and learnable weights $w \in \mathcal{W}$, where $\mathcal{A}$ is the architecture space and $\mathcal{W}$ is the weight space:

$$\alpha^* = \arg \min_{\alpha \in \mathcal{A}} \mathcal{L}(F_{\alpha, w^*(\alpha)}(\mathcal{G}), \mathcal{Y}), \quad (2)$$

$$\text{s.t.} \quad w^*(\alpha) = \arg \min_{w \in \mathcal{W}(\alpha)} \mathcal{L}(F_{\alpha, w}(\mathcal{G}), \mathcal{Y}), \quad (3)$$

where $\mathcal{L}$ represents the loss of predictions made by the architecture $F_{\alpha, w}(\cdot)$ on the graph, and $\alpha^*$ and $w^*$ denote the optimal architecture and weights for the given data $\mathcal{D} = (\mathcal{G}, \mathcal{Y})$. Specifically, $\alpha$ typically represents the selection of GNN operations (e.g., GCN (Kipf & Welling, 2017), GraphSAGE (Hamilton et al., 2017), GAT (Velickovic et al., 2017), GIN (Xu et al., 2018), etc.), which are referred to as operation choices for brevity. GNAS addresses this as a bi-level optimization problem (Elsken et al., 2019).

### 2.3. GNAS for GNN-based GFMs

We define diverse data as $\mathcal{D} = \{\mathcal{D}_1, \mathcal{D}_2, \ldots, \mathcal{D}_N\}$, where $\mathcal{D}_i = \{\mathcal{G}_i, \mathcal{Y}_i\}$ represents the $i$-th dataset with graph $\mathcal{G}_i$ and label $\mathcal{Y}_i$. Following previous work for GNN-based GFMs (Liu et al., 2023a; Wang et al., 2024b), which leverage

LLMs to unify the node feature space across different graphs and leverage subgraphs to unify graph tasks, enabling a single GNN to be applied to diverse data $\mathcal{D}$ across domains and tasks. Graph neural architecture search for GFM aims to search a graph neural architecture $F_{\alpha,w}$ that achieves performance in diverse data $\mathcal{D}$.

## 3. Preliminaries

In this section, we first introduce the problem of *architecture inconsistency* in GFM. Then we provide an invariant view of architecture customization and formulate the overall objective for our proposed method.

### 3.1. Architecture Inconsistency in GFM

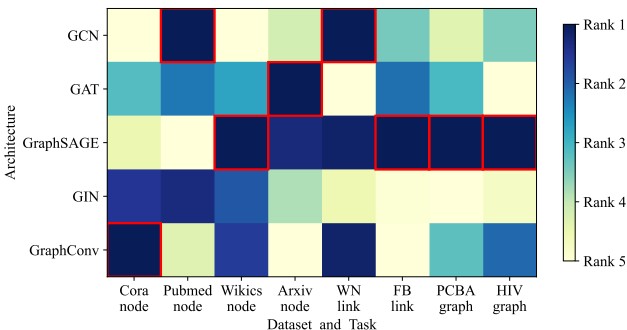

*Figure 1.* A heatmap visualization illustrating that the optimal architecture can vary across datasets with different domains and tasks. The darker the color of a block, the better the performance on the corresponding dataset. Additionally, red indicates the best-performing architectures for each dataset.

According to our observation, the optimal architecture for graph data across different tasks and domains may exhibit *architecture inconsistency*, meaning that the optimal architectures vary for data with diverse tasks and domains. To validate this, we test various GNN architectures built upon a GNN-based GFM, GFT (Wang et al., 2024b), on datasets with different domains and tasks. We present the performance of each architecture on each dataset using a heatmap in Figure 1. As shown in Figure 1, datasets from different domains and tasks require distinct optimal architectures, highlighting the presence of *architecture inconsistency*. Based on this observation, we introduce the following assumption:

**Assumption 3.1.** There exist two datasets $\mathcal{D}_i, \mathcal{D}_j \in \mathcal{D}$, the optimal operation required $\mathcal{D}_i$ is different from $\mathcal{D}_j$.

Assumption 3.1 assumes that the optimal architectures required by two different datasets may differ. Then we further provide theoretical analyses showing that *architecture inconsistency* leads to operations optimization conflicts in existing differentiated GNAS methods. We have the follow-

ing proposition with proof in Appendix A.1.

**Proposition 3.2.** *If there exist two datasets $\mathcal{D}_i, \mathcal{D}_j \in \mathcal{D}$, the optimal operation for $\mathcal{D}_i$ is different from $\mathcal{D}_j$, the operations will render optimization conflicts.*

Assumption 3.1 serves as a prerequisite condition for Proposition 3.2. Proposition 3.2 demonstrates that when two datasets require different optimal architectures, current mainstream GNAS methods encounter optimization conflicts for GFM. For instance, as illustrated in Figure 1, the optimal architectures for PubMed and Wikics differ. When existing GNAS methods search simultaneously for an architecture optimal for both datasets, they fail to identify a single architecture that performs best for both and are forced to compromise.

To tackle *architecture inconsistency*, our key idea is to train a mapping function $\pi: \mathcal{G} \rightarrow \mathcal{A}$, which customizes architectures for each data to prevent *architecture inconsistency*, while simultaneously facilitating knowledge sharing across domains and tasks via the weight-sharing supernetwork.

### 3.2. Invariant View of Architecture Customization

We customize graph neural architectures for graph data across diverse tasks and domains from an invariant perspective. Unlike conventional invariant inference approaches that aim to discover invariant relationships between data and labels (Wu et al., 2022b; Li et al., 2022a; Wu et al., 2022a), our goal is to identify invariant relationships between the graph data and the corresponding architecture, addressing the issue of *architecture inconsistency* by tailoring graph neural architectures for each data individually.

We formalize the four key variables: input graph data $G$, architecture $A$, invariant pattern $Z_I$, and variant pattern $Z_V$. We divide the architecture mapping function $\pi$ into two components: encoder $\theta: G \rightarrow Z_I$ and predictor $\psi: Z_I \rightarrow A$. We make the following assumptions:

**Assumption 3.3.** (1) $Z_I = G \setminus Z_V$. There are two disjoint parts in the graph data $G$: invariant part $Z_I$ and variant part $Z_V$. (2) $Z_V \not\perp A$. The variant part $Z_V$ is correlated with the architecture $A$. (3) $A \perp Z_V \mid Z_I$ and $A = \psi(Z_I)$. $Z_I$ shields $A$ from the influence of $Z_V$.

Assumption 3.3 defines what constitutes an invariant pattern for architecture prediction: i) **Condition 1** indicates that the data contains two types of patterns: an invariant pattern $Z_I$, which reliably predicts the architecture, and a variant pattern $Z_V$, which cannot stably predict the architecture; ii) **Condition 2** highlights that the variant pattern $Z_V$ is not independent of the architecture $A$; iii) **Condition 3** states that, given the invariant pattern $Z_I$, the architecture $A$ is independent of the variant pattern $Z_V$, and $Z_I$ is sufficient for predicting $A$.

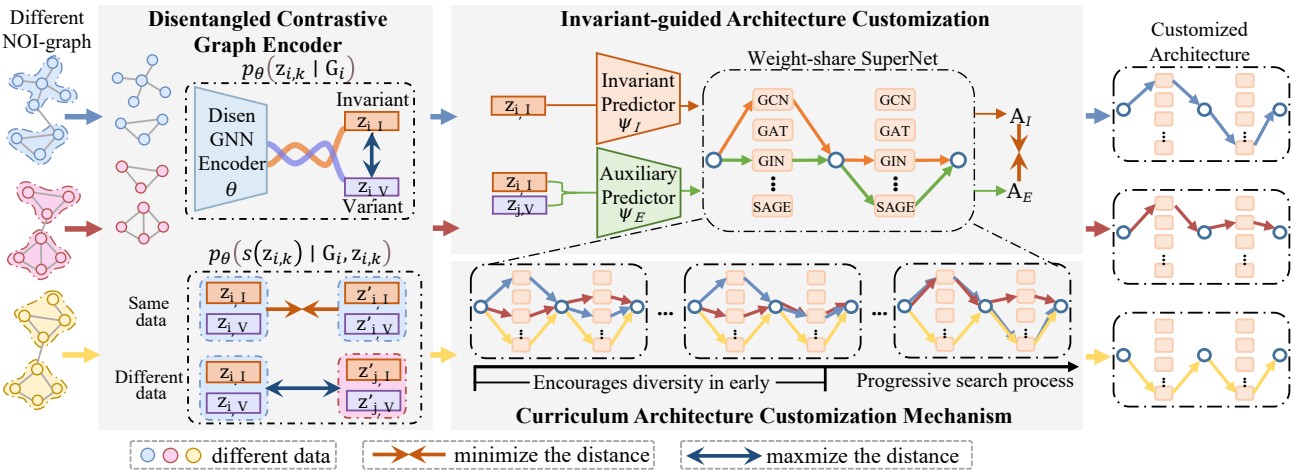

*Figure 2.* The framework of Automated Graph Foundation Model with Adaptive Architecture Customization(**AutoGFM**). The model consists of three modules: i) Disentangled contrastive graph encoder that discovers invariant and variant patterns from graph data , ii) invariant-guided architecture customization enabling customization of graph neural architectures based on the discovered invariant and variant patterns to adapt to data with diverse domains and tasks, and iii) Curriculum architecture customization mechanism to mitigate the influence of any single data dominating the search process.

## 3.3. Overall Objective

To satisfy the constraints outlined in Assumption 3.3, we formulate a learning objective that adheres to the specified conditions. Specifically, we minimize the mutual information between $Z_I$ and $Z_V$ to ensure that the two parts remain disjoint. Simultaneously, we maximize the mutual information between $Z_I$ and $A$ , ensuring that $Z_I$ is sufficient for predicting the architecture $A$. Furthermore, we minimize the mutual information between $A$ and $Z_V$, conditioned on $Z_I$, to guarantee that $Z_I$ shields $A$ from the influence of $Z_V$. The resulting overall learning objective is defined as follows:

$$\max_{\theta,\psi} \; I(Z_I, A) - \lambda I(Z_I, Z_V) - \beta I(A, Z_V \mid Z_I), \quad (4)$$

where $I$ denotes the mutual information function, $\theta$ represents the encoder that extracts $Z_I$ and $Z_V$ from $G$, $\psi$ represents the predictor that maps $Z_I$ to $A$, and $\lambda$ and $\beta$ are hyperparameters controlling the trade-off.

## 4. The Proposed Method: AutoGFM

In this section, we introduce an Automated Graph Foundation Model with Adaptive Architecture Customization (**AutoGFM**) to search for graph neural architectures for each graph data with diverse tasks and domains individually. We first introduce two modules: a disentangled contrastive graph encoder and invariant-guided architecture customization. Besides, we introduce our optimization objective with a curriculum architecture customization mechanism. The overall framework of **AutoGFM** is illustrated in Figure 2.

## 4.1. Disentangled Contrastive Graph Encoder

In this section, we focus on learning disentangled representations to capture two distinct aspects of graph data. Specifically, we aim to learn two architecture-aware disentangled representations, $Z_I$ and $Z_V$, we temporarily treat them jointly and denote the two-channel representations as $Z_k$ ($k = 1, 2$) in this section. The main insight of our proposed method is intuitively based on the following observations: (1) Data from the same sources (i.e., the same domain and task) require similar architectures, so they share a similar $Z_I$ that reflects the architectural requirements. Conversely, graphs from different data sources will have distinct $Z_I$. (2) To satisfy the Assumption 3.3, the mutual information between $Z_I$ and $Z_V$ should be minimized.

**Disentangled NOI-graph Encoder** Initially, we adopt GNNs with individual parameters to learn two-channel graph representations of NOI-graphs.

$$H_k^{(l)} = \text{GNN}_k\big(H_k^{(l-1)}, \mathbf{A}\big), \quad k = 1, 2, \quad (5)$$

where $H_k^{(l)}$ is the $k$-th channel of the node representation at the $l$-th layer, $\mathbf{A}$ is the adjacency matrix of the graph. We employ two distinct Readout functions (i.e., pooling functions) and MLPs to derive a NOI-graph-level representation for each channel:

$$z_k = \text{MLP}_k\big(h_k\big), \quad (6)$$
$$h_k = \text{Readout}_k\big(H_k^{(L)}\big), \quad k = 1, 2. \quad (7)$$

**NOI-graph Disentangled Contrastive Learning** Inspired by self-supervised contrastive learning that captures

discriminative features by pulling similar samples together and pushing dissimilar samples apart in latent space (Jaiswal et al., 2020; Le-Khac et al., 2020; You et al., 2020), we propose an NOI-graph-level contrastive learning method to encourage disentangled representations to reflect the architectural requirements of the graph data. Initially, we encourage the representations $Z_I$ and $Z_V$ to be disentangled:

$$p_\theta\left(z_{i,k} \mid G_i\right) = \frac{\exp \phi\left(z_{i,k}, p_k\right)}{\sum_{j=1}^{2} \exp \phi\left(z_{i,k}, p_j\right)}, \qquad (8)$$

where $\phi$ is a similarity function, $G_i$ is a NOI-graph from $i$-th graph, $z_{i,k}$ is the $k$-th chunk of the representation of a NOI-graph from $i$-th graph, and $p_k$ is the propotype of the $k$-th chunk of the representation. Then, we propose a NOI-graph-level instance discriminative task to encourage the representations $Z_I$ to capture different architectural requirements of different data. The task is defined as:

$$p_\theta\left(s(z_{i,k}) \mid G_i, z_{i,k}\right) = \frac{\exp \phi\left(z_{i,k}, z'_{i,k}\right)}{\sum_{j=1}^{N} \exp \phi\left(z_{i,k}, z'_{j,k}\right)}, \quad (9)$$

where $s(z_{i,k})$ represents a unique surrogate label assigned to $z_{i,k}$, and $z'_{i,k}$ is sampled from the same graph data $G_i$ as $z_{i,k}$. Then we learn the model parameters $\theta$ by calculating the loss function as:

$$\mathcal{L}_{dis} = \sum_i -\log \mathbb{E}_{p_\theta(z_{i,k}|G_i)} p_\theta\left(s(z_{i,k}) \mid G_i, z_{i,k}\right). \quad (10)$$

In this way, we encourage the representations $Z_I$ to capture the architectural requirements of the graph data, while ensuring that the representations $Z_I$ and $Z_V$ are disentangled.

### 4.2. Invariant-guided Architecture Customization

To customize graph neural architectures based on the discovered patterns and enable adaptation to data from diverse domains and tasks, we propose an invariant-guided architecture customization approach. Specifically, we first establish a weight-sharing super-network with a set of prototypes, then utilize an invariant predictor, $\psi_I$, and an auxiliary predictor, $\psi_E$, to guide the customization process, ensuring the minimization of $I(A, Z_V \mid Z_I)$ in Assumption 3.3.

**Weight-sharing Super-network** To facilitate differentiable optimization, we employ continuous parameterization and a weight-sharing mechanism (Liu et al., 2018) to implement the mixed operations. The super-network layer with $|\mathcal{O}|$ mixed operations is defined as:

$$H^{(l)} \leftarrow \sum_{i=1}^{|\mathcal{O}|} \alpha_{l,i} GNN_i^{(l-1)}(H^{(l-1)}, A), \qquad (11)$$

where $A$ is the adjacency matrix of the graph, $H^{(l)}$ represents the node representations at the $l$-th layer, $GNN_i^{(l-1)}$ denotes the mixed GNN operations, $|\mathcal{O}|$ is the number of GNN operation choices, and $\alpha_{l,i}$ indicates the probability of selecting the $i$-th operation for the $l$-th layer. Different from previous super-networks (Liu et al., 2018) that use learnable parameters $\alpha$, we employ a set of prototypes to guide the routing between data and the operations.

**Architecture Predictor** Given a graph representation $z \in Z$, we design an invariant mapping predictor, $\psi_I$: $z \to \{\alpha_{l,i}\}$. The probability $\alpha_{l,i}$ of selecting the $i$-th operation for the $l$-th layer is calculated as follows:

$$\alpha_{l,i} = \frac{\exp(\hat{\alpha}_{l,i})}{\sum_{j=1}^{|\mathcal{O}|} \exp(\hat{\alpha}_{l,j})}, \quad \hat{\alpha}_{l,i} = z \cdot \frac{p_{l,i}}{\|p_{l,i}\|_2}, \qquad (12)$$

where $p_{l,i}$ is a learnable prototype of the $i$-th operation for the $l$-th layer, and $z$ is the graph representation. Following (Qin et al., 2022a), we adopt the $l_2$-normalization on $p$ to ensure numerical stability and fair competition among different operations. We utilize the learnable prototypes $p$ as the parameters of the predictor $\psi_I$ to map architecture, *i.e.*, if the graph representation $z$ is similar to the prototype $p_{l,i}$, the operation $i$ will be selected for the $l$-th layer.

**Invariant-guided Customization** The objective of minimizing the conditional mutual information $I(A, Z_V \mid Z_I)$ in Equation (4) is not tractable, as the mutual information of high-dimensional vectors is difficult to estimate. Therefore, we utilize an equivalent transformation in Proposition 4.1 to achieve it, with a detailed proof provided in Appendix A.2.

**Proposition 4.1.** *if* $P(A \mid Z_I, Z_V) = P(A \mid Z_I)$, *the conditional mutual information* $I(A, Z_V \mid Z_I)$ *achieves its minimum value of* 0.

Proposition 4.1 indicates that we can minimize the conditional mutual information $I(A, Z_V \mid Z_I)$ by enforcing $P(A \mid Z_I, Z_V) = P(A \mid Z_I)$. To this end, we utilize an auxiliary predictor $\psi_E$ with an invariant predictor $\psi_I$ to guide the customization process. Specifically, we first predict the architecture $A_I$ based on the invariant patterns $Z_I$ using the invariant predictor $\psi_I$. Then, we predict the architecture $A_E$ based on the patterns fused from $Z_I$ and $Z_V$ using the auxiliary predictor $\psi_E$:

$$A_I = \psi_I(Z_V), \quad A_E = \psi_E(Z_I, Z_V). \qquad (13)$$

We guide $P(A \mid Z_I, Z_V) = P(A \mid Z_I)$ by minimizing the difference between the two predictions $A_I, A_E$. Besides, to boost the architecture predictor to fit more data with different variant patterns, we fuse $Z_I$ with $Z_V$ from other

data to predict $A_E$. We define the loss as:

$$\mathcal{L}_{\text{inv}} = \sum_i^{\|\mathcal{D}\|} \sum_j^{\|\mathcal{D}\|} \|A_{I,i} - A_{E,(i,j)}\|, \quad (14)$$

$$s.t. \quad A_{I,i} = \psi_I(z_{I,j}), A_{E,i,j} = \psi_E(z_{I,i}, z_{V,j}), \quad (15)$$

where $A_{I,i}$ is the predicted architecture for the $i$-th graph based on the invariant patterns $z_I$, $A_{E,i,j}$ are the predicted architectures based on the patterns fused from $z_{I,i}$ and $z_{V,j}$.

## 4.3. Optimization with Curriculum Customization Mechanism

We calculate the task loss of GFM using only the architecture predicted by the invariant predictor $\psi_I$:

$$\mathcal{L}_{task} = \ell(F_{\psi(Z_I)}(G), y). \quad (16)$$

$\mathcal{L}_{task}$ aim to maximize $I(Z_I, A)$ in Equation (4). Notably, the computation method of $\mathcal{L}_{task}$ depends on the GFM for which we aim to search for architectures. *e.g.*, GFT (Wang et al., 2024b) utilizes Computation Tree Reconstruction to calculate loss during the pretraining stage.

GFMs need to be simultaneously optimized using multiple datasets with diverse domains and tasks. However, different data have different influences on architectures (Zhou et al., 2022d), mainly on the learnable weights of operations in our study, *e.g.*, some datasets are easier to fit with certain operations but more challenging with others. As a result, operations that fit well in the early stages of training are more likely to be selected, causing other operations to be overlooked.

To mitigate the dominance of data in the search process, we design a curriculum architecture customization constraint that encourages diversity in the customized architectures during the early stages of training. We first calculate the average $\alpha$ for each operation in the $l$-th layer as:

$$\boldsymbol{\alpha}_l = \frac{\left[\sum_{i=1}^{|\mathcal{D}|} \alpha_{l,1}(z_i), \sum_{i=1}^{|\mathcal{D}|} \alpha_{l,2}(z_i), \dots, \sum_{i=1}^{|\mathcal{D}|} \alpha_{l,J}(z_i)\right]}{|\mathcal{D}|}, \quad (17)$$

where $\boldsymbol{\alpha}_l$ is the average $\alpha$ for each operation in the $l$-th layer, $\alpha_{l,j}(z_i)$ is the probability of selecting the $j$-th operation for the $l$-th layer predicted by the invariant predictor $\psi_I$ for the $i$-th graph, and $J$ is the number of operations. We define the curriculum architecture customization loss as follows:

$$\mathcal{L}_{cur} = \gamma \sum_{l=1}^{L} \text{CV}(\boldsymbol{\alpha}_l), \quad (18)$$

where $\text{CV}(\boldsymbol{\alpha}_l)$ is the coefficient of variation of the average $\alpha$ for each operation in the $l$-th layer across different data and

---

**Algorithm 1** Training pipeline for **AutoGFM**

---
**Input:** data $\mathcal{D} = \{\mathcal{G}_1, \mathcal{G}_2, \dots, \mathcal{G}_N\}$, hyperparameters $\lambda, \beta$.
**for** $t = 1, \dots, T$ **do**
    Sample NOI-graphs $G_i$ from $\mathcal{G}_i$.
    Extract $Z_I$ and $Z_V$ from NOI-graphs $G_i$.
    Calculate $\mathcal{L}_{dis}$ using Equation (10).
    Obtain architectures $A_I$ and $A_E$ predicted by $\psi_I$ and $\psi_E$ in Equation (15).
    Calculate $\mathcal{L}_{inv}$ using Equation (14).
    Calculate $\mathcal{L}_{cur}$ and $\mathcal{L}_{task}$ using Equation (18) and Equation (16), respectively.
    Update $\theta, \psi_I, \psi_E$ by minimizing Equation (19).
**end for**

---

$\gamma$ is controlled by a pacing function: $\gamma = 1 - \frac{t}{t_e}$, where $t$ is the current training step and $t_e$ is the step to stop the curriculum customization mechanism. This constraint encourages diversity in early architecture customization, thereby mitigating the influence of any single data dominating the search process. The final training objective is:

$$\min_{\theta, \psi_I, \psi_E} \overset{\max I(Z_I, A)}{\underset{\min I(Z_I, Z_V)}{\mathcal{L}_{task}}} + \lambda \mathcal{L}_{dis} + \beta \overset{\min I(A, Z_V \mid Z_I)}{\mathcal{L}_{inv}} + \mathcal{L}_{cur}, \quad (19)$$

where $\mathcal{L}_{task}$ aims to exploit invariant patterns to customize architectures, $\mathcal{L}_{dis}$ encourages the disentanglement of the invariant and variant patterns, $\mathcal{L}_{inv}$ discovers the invariant patterns and variant patterns, and $\mathcal{L}_{cur}$ mitigates the influence of any single data dominating the search process. The overall algorithm is summarized in Algorithm 1.

During the inference stage, given an input graph, we first utilize the disentangled contrastive graph encoder to obtain its invariant pattern representation, denoted as $Z_I$. Then, $Z_I$ is fed into the invariant predictor $\psi_I$ to generate a customized architecture. This architecture is then used as the GNN component within the GFM to perform prediction.

## 5. Experiments

In this section, we conduct experiments on real-world datasets with diverse domains and tasks to show the effectiveness of the proposed **AutoGFM** for GFM.

### 5.1. Experimental Setup

**Datasets** We employ datasets with diverse domains and tasks. For node-level tasks, we utilize citation networks (Cora, Pubmed, and Arxiv) and the web link network (WikiCS). For edge-level tasks, we utilize Knowledge Graphs (WN18RR, FB15K237). For graph-level tasks, we utilize molecular datasets (HIV, PCBA, and ChEMBL). Following

*Table 1.* Accuracy (%) with std of different methods in pre-training and fine-tuning setting. The highest result is **bold**. The subscript v represents Vanilla GNNs. We do not explicitly report the performance of GFT separately, as GFT (Wang et al., 2024b) employs GraphSAGE as its GNN architecture, which overlaps with our baselines.

| Method | Node Classification | | | | Link Classification | | Graph Classification | | *Avg.* |
|---|---|---|---|---|---|---|---|---|---|
| | Cora | PubMed | Wiki-CS | Arxiv | WN18RR | FB15K237 | HIV | PCBA | |
| Linear | $58.03_{\pm2.33}$ | $68.66_{\pm2.24}$ | $70.36_{\pm0.58}$ | $66.50_{\pm0.14}$ | $78.50_{\pm0.59}$ | $87.39_{\pm0.07}$ | $66.37_{\pm1.11}$ | $72.30_{\pm0.34}$ | 71.01 |
| GCN$_v$ | $75.65_{\pm1.37}$ | $75.61_{\pm2.10}$ | $75.28_{\pm1.34}$ | $71.40_{\pm0.08}$ | $73.79_{\pm0.39}$ | $82.22_{\pm0.28}$ | $64.84_{\pm4.78}$ | $71.32_{\pm0.49}$ | 73.76 |
| GAT$_v$ | $76.24_{\pm1.62}$ | $74.86_{\pm1.87}$ | $76.78_{\pm0.78}$ | $70.87_{\pm0.24}$ | $80.16_{\pm0.27}$ | $88.93_{\pm0.15}$ | $65.54_{\pm6.93}$ | $70.12_{\pm0.89}$ | 75.44 |
| GIN$_v$ | $73.59_{\pm2.10}$ | $69.51_{\pm6.87}$ | $49.77_{\pm4.72}$ | $65.05_{\pm0.50}$ | $74.02_{\pm0.55}$ | $83.21_{\pm0.53}$ | $66.86_{\pm3.48}$ | $72.69_{\pm0.22}$ | 69.34 |
| DGI | $72.10_{\pm0.34}$ | $73.13_{\pm0.64}$ | $75.32_{\pm0.95}$ | $69.15_{\pm0.20}$ | $75.75_{\pm0.59}$ | $81.34_{\pm0.15}$ | $59.62_{\pm1.21}$ | $63.31_{\pm0.89}$ | 71.22 |
| BGRL | $71.20_{\pm0.30}$ | $75.29_{\pm1.33}$ | $76.53_{\pm0.69}$ | $71.19_{\pm0.18}$ | $75.44_{\pm0.30}$ | $80.66_{\pm0.29}$ | $63.95_{\pm1.06}$ | $67.09_{\pm1.00}$ | 72.67 |
| GraphMAE | $73.10_{\pm0.40}$ | $74.32_{\pm0.33}$ | $77.61_{\pm0.39}$ | $70.90_{\pm0.31}$ | $78.99_{\pm0.48}$ | $85.30_{\pm0.16}$ | $61.04_{\pm0.55}$ | $63.30_{\pm0.78}$ | 73.07 |
| GIANT | $75.13_{\pm0.49}$ | $72.31_{\pm0.53}$ | $76.56_{\pm0.88}$ | $70.10_{\pm0.32}$ | $84.36_{\pm0.30}$ | $87.45_{\pm0.54}$ | $65.44_{\pm1.39}$ | $61.49_{\pm0.99}$ | 74.11 |
| GCN | $77.97_{\pm1.46}$ | $77.68_{\pm1.43}$ | $76.35_{\pm0.50}$ | $67.22_{\pm0.80}$ | $92.20_{\pm0.40}$ | $77.35_{\pm3.62}$ | $70.89_{\pm4.52}$ | $74.94_{\pm1.69}$ | 76.83 |
| GAT | $78.96_{\pm0.91}$ | $77.24_{\pm2.04}$ | $78.00_{\pm0.67}$ | $72.82_{\pm0.33}$ | $75.91_{\pm1.29}$ | $86.15_{\pm2.17}$ | $69.07_{\pm1.22}$ | $76.23_{\pm0.61}$ | 76.80 |
| GraphSAGE | $78.24_{\pm1.46}$ | $76.28_{\pm2.19}$ | $79.29_{\pm0.53}$ | $72.28_{\pm0.24}$ | $91.57_{\pm0.42}$ | $89.92_{\pm0.27}$ | $72.85_{\pm2.47}$ | $78.32_{\pm0.24}$ | 79.84 |
| GIN | $79.85_{\pm1.30}$ | $77.57_{\pm2.04}$ | $78.61_{\pm0.55}$ | $67.80_{\pm0.52}$ | $77.87_{\pm1.12}$ | $71.37_{\pm3.06}$ | $69.32_{\pm2.84}$ | $74.27_{\pm1.70}$ | 74.58 |
| GraphConv | $80.12_{\pm1.27}$ | $76.52_{\pm1.39}$ | $78.85_{\pm0.66}$ | $65.75_{\pm0.60}$ | $91.52_{\pm0.33}$ | $80.72_{\pm2.05}$ | $71.81_{\pm2.59}$ | $76.06_{\pm0.46}$ | 77.67 |
| Darts | $76.97_{\pm1.34}$ | $77.77_{\pm1.59}$ | $73.60_{\pm2.20}$ | $72.10_{\pm1.64}$ | $78.02_{\pm1.52}$ | $81.64_{\pm3.00}$ | $68.85_{\pm3.25}$ | $75.16_{\pm2.21}$ | 75.51 |
| Graphnas | $76.34_{\pm2.25}$ | $77.49_{\pm2.02}$ | $70.98_{\pm2.22}$ | $68.64_{\pm2.04}$ | $82.63_{\pm0.44}$ | $80.72_{\pm3.65}$ | $67.70_{\pm4.50}$ | $73.49_{\pm2.82}$ | 74.75 |
| GASSO | $78.24_{\pm1.50}$ | $77.82_{\pm1.68}$ | $71.90_{\pm2.00}$ | $70.85_{\pm1.90}$ | $81.96_{\pm0.58}$ | $80.66_{\pm2.21}$ | $68.37_{\pm6.28}$ | $76.76_{\pm1.64}$ | 75.82 |
| Graces | $78.30_{\pm1.92}$ | $76.98_{\pm2.57}$ | $70.05_{\pm3.06}$ | $70.23_{\pm1.22}$ | $84.05_{\pm0.49}$ | $83.91_{\pm3.99}$ | $70.93_{\pm2.24}$ | $75.89_{\pm1.54}$ | 76.29 |
| Ours | $\mathbf{80.32_{\pm1.12}}$ | $\mathbf{78.28_{\pm1.40}}$ | $\mathbf{79.45_{\pm0.69}}$ | $\mathbf{73.39_{\pm1.56}}$ | $\mathbf{93.17_{\pm0.88}}$ | $\mathbf{90.27_{\pm1.64}}$ | $\mathbf{73.17_{\pm2.21}}$ | $\mathbf{78.83_{\pm1.54}}$ | **80.86** |

(Liu et al., 2023a), we use the textual encoder to unify the node features from different domains.

**Baselines** We compare our proposed **AutoGFM** with the five categories of baselines: (1) **Vanilla GNNs**: GCN (Kipf & Welling, 2017), GAT (Velickovic et al., 2017), GIN (Xu et al., 2018); (2) **Self-supervised methods**: BGRL (Thakoor et al., 2021), GraphMAE (Hou et al., 2022), GIANT (Chien et al., 2022). (3) **GFMs**: OFA (Liu et al., 2023a) and GFT (Wang et al., 2024b). (4) **Manually designed GNNs**: GCN (Kipf & Welling, 2017), GAT (Velickovic et al., 2017), GIN (Xu et al., 2018), GraphSAGE (Hamilton et al., 2017), and GraphConv (Morris et al., 2019). (5) **GNAS methods**: DARTS (Liu et al., 2018), GraphNAS (Gao et al., 2021), GASSO (Qin et al., 2021b), Graces (Qin et al., 2022a).

For manually designed GNNs, GNAS baselines, and **AutoGFM** , we utilize GFT (Wang et al., 2024b) as the base model to ensure a fair comparison. Additionally, we adopt the same search space (operations in manually designed GNNs baselines and super-network layers is 2) for both GNAS baselines and **AutoGFM**. We replicate each experiment ten times and report the average results. Further details about experimental setups are provided in Appendix D.

### 5.2. Main Results

**Pre-training and Fine-tuning** From the results in Table 1, we observe the following: (1) None of manually designed GNN performs well across all datasets, indicating that fixed architectures struggle to generalize across diverse domains and tasks. (2) Existing GNAS methods fail to discover better architectures for each dataset, highlighting their limitations in adapting to diverse domains and tasks. (3) **AutoGFM** outperforms all baselines across datasets, demonstrating its effectiveness in customizing architectures for different domains and tasks.

**Few-shot Learning** Few-shot learning is a challenging task that requires models to generalize well with limited labeled samples. We randomly sample a few labeled samples per way from the training set for fine-tuning. From the results in Table 2, despite the limited labeled samples, **AutoGFM** achieves the best performance across all datasets, demonstrating the fast adaptability of its architectures. We provide more experimental results in Appendix B.3.

### 5.3. Ablation Study

To verify the effectiveness of the key modules in our method, we compare different ablated versions on five datasets: i) w/o D removes and replaces the disentangled contrastive graph encoder with standard GNNs; ii) w/o I removes the invariant-guided architecture customization module by removing the $\mathcal{L}_{inv}$; iii) w/o C removes the curriculum architecture customization mechanism. The results are shown in Figure 3. We observe that the full model achieves the best performance across all datasets, demonstrating the ef-

*Table 2.* Accuracy (%) with std of different methods in Few-shot learning. The highest result is **bold**.

| Method | Cora-7 way | | | WN18RR-10 way | | | CHEMHIV-2 way | | | |
| | 1-shot | 3-shot | 5-shot | 1-shot | 3-shot | 5-shot | 1-shot | 3-shot | 5-shot | 10-shot |
| --- | --- | --- | --- | --- | --- | --- | --- | --- | --- | --- |
| OFA | 30.38±2.39 | 36.03±2.11 | 32.10±1.79 | 25.82±1.07 | 30.56±1.02 | 32.64±1.56 | 57.17±1.82 | 59.30±3.04 | 57.56±3.66 | 54.36±4.90 |
| GFT | 41.40±8.04 | 43.31±8.11 | 43.55±7.43 | 35.33±4.20 | 35.50±5.02 | 35.50±4.59 | **59.94±7.09** | 58.44±7.28 | 58.78±6.92 | 58.67±7.54 |
| GCN | 43.07±7.37 | 42.38±7.42 | 42.57±7.50 | 29.85±4.14 | 29.78±3.64 | 30.40±3.02 | 59.58±6.15 | 59.53±8.21 | 59.28±6.79 | 59.64±7.82 |
| GAT | 46.12±7.10 | 47.31±7.78 | 47.71±8.02 | 34.50±2.98 | 34.37±3.43 | 34.70±3.15 | 56.39±9.80 | 59.17±9.43 | 59.33±7.57 | 59.22±7.20 |
| GraphSAGE | 40.50±6.11 | 42.07±6.12 | 42.40±6.12 | 38.03±2.03 | 38.17±2.34 | 38.30±2.16 | 58.33±4.28 | 58.64±6.22 | 59.28±6.96 | 58.17±9.16 |
| GIN | 45.29±6.26 | 47.02±7.32 | 47.24±7.33 | 36.62±4.17 | 36.92±4.03 | 37.47±3.10 | 58.72±4.45 | 57.97±3.73 | 59.06±4.17 | 57.22±4.89 |
| GraphConv | 38.67±8.50 | 40.93±8.80 | 41.60±9.20 | 38.93±3.77 | 39.28±2.27 | 39.62±3.44 | 53.00±7.75 | 55.50±8.45 | 54.67±8.17 | 53.06±7.47 |
| DARTS | 43.29±7.65 | 42.10±7.45 | 42.81±7.92 | 37.22±2.68 | 38.57±3.22 | 38.65±3.55 | 58.31±6.73 | 58.68±6.72 | 58.07±6.17 | 59.05±8.83 |
| GraphNAS | 38.64±8.31 | 40.60±9.23 | 41.62±9.20 | 36.79±3.17 | 37.03±3.32 | 36.98±5.23 | 57.62±5.70 | 58.52±7.05 | 58.91±6.14 | 59.32±7.75 |
| GASSO | 40.31±6.10 | 42.07±5.85 | 42.95±5.75 | 37.13±3.52 | 37.42±1.91 | 37.37±3.31 | 59.46±5.35 | 59.38±7.10 | 59.26±6.45 | 59.72±5.42 |
| GRACES | 45.43±6.73 | 46.31±7.42 | 47.57±7.22 | 38.76±2.63 | 38.24±3.60 | 39.13±2.29 | 59.39±4.01 | 58.62±9.30 | 59.45±5.79 | 59.71±7.18 |
| Ours | **46.29±7.24** | **47.33±7.80** | **47.76±8.06** | **39.34±3.03** | **39.55±2.46** | **40.02±2.26** | 59.73±4.46 | **59.68±7.74** | **60.08±4.22** | **59.92±4.08** |

fectiveness of each module. We further observe the following: i) The disentangled contrastive graph encoder module is designed to extract discriminative invariant and variant patterns from the data by pulling similar samples closer and pushing dissimilar samples apart in the latent space. Removing this module impairs the extraction of invariant patterns and reduces the distinguishability between patterns extracted from different datasets, ultimately harming the effectiveness of architecture prediction; ii) The invariant-guided architecture customization module serves to shield architecture A from the influence of variant patterns $Z_V$ given the invariant pattern $Z_I$. The substantial performance decrease observed upon removing this module highlights the importance of effectively isolating architecture predictions from $Z_V$ influences, reinforcing the critical role of this module in ensuring the invariance conditions of captured patterns; iii) This curriculum architecture customization mechanism aims to reduce data dominance in the architecture search process. Removing this module causes certain operations, which perform well on specific datasets during early training stages, to dominate the search process. Consequently, other datasets may neglect potentially beneficial operations.

## 5.4. Time Complexity Analysis

Let $|V|$ and $|E|$ denote the number of nodes and edges, and $d$ as the dimensionality. We use $d_e$ and $d_a$ to denote the dimensionality of the disentangled graph encoder and the customized super-network. The time complexity of the GNN layers in both the graph encoder and the super-network is $O(|E|d + |V|d^2)$. Therefore, the time complexity of our disentangled graph encoder is $O(|E|d_e + |V|d_e^2)$. The time complexity of the architecture customization with prototypes is $O(|\mathcal{O}|^2 d_e)$. The time complexity of the customized super-network is $O(|\mathcal{O}|(|E|d_a + |V|d_a^2))$. Thus, the over-

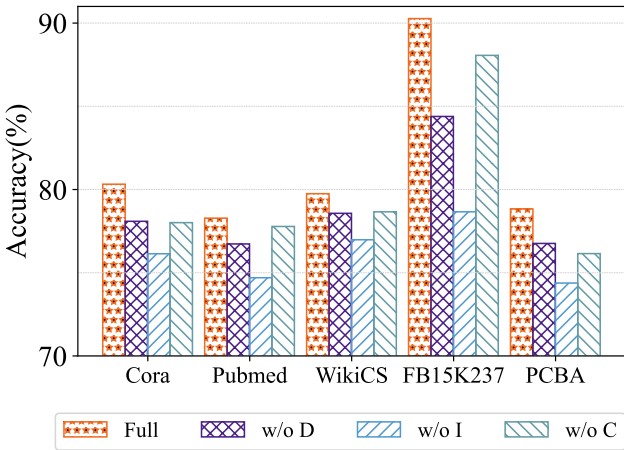

*Figure 3.* Comparisons of different ablated versions of **AutoGFM** on real-world datasets. "Full" denotes the full version of the method.

all computational complexity of our method is given by:
$$O(|E|d_e + |V|d_e^2 + |\mathcal{O}|^2 d_e + |\mathcal{O}|(|E|d_a + |V|d_a^2)).$$

## 6. Related Work

In this section, we review the related work on GNN-based Graph Foundation Models, graph neural architecture search, graph invariant presentation learning, and graph self-supervised learning. We provide more related work in Appendix C.

### 6.1. GNN-based Graph Foundation Models

GNN-based GFMs, which leverage LLMs as enhancers, are a promising direction for GFM (He & Hooi, 2024; Mao et al., 2024a; Zhao et al., 2024b; Xia & Huang, 2024; Huang

et al., 2024c). Specifically, it involves using LLMs to transform the textual features of graphs into unified representations and unify graph-related tasks through subgraph classification for GNNs (Sun et al., 2023; Zhao et al., 2024c; Fan et al., 2024; Ren et al., 2024; Pan et al., 2024b; Mao et al., 2024b). This enables the transfer of shared knowledge across various domains. Two key challenges in developing GNN-based Graph Foundation Models (GFMs) are the unification of diverse tasks and domain spaces (Yu et al., 2024; Li et al., 2024b; Zhao et al., 2024a; Xia et al., 2024; Zhu et al., 2024b; Guo et al., 2023). For instance, OFA (Liu et al., 2023a) employs an LLM to unify input features from diverse datasets, converting multiple graph classification tasks into a unified binary classification format. GFT (Wang et al., 2024b) extends OFA by incorporating computation trees to discover transferable patterns across graphs. Nevertheless, these models face challenges in their reliance on manually designed architectures, which can constrain their performance on diverse domains and tasks. More related work about GFMs is included in Appendix C.

### 6.2. Graph Neural Architecture Search

Neural architecture search (NAS) has gained growing attention for its capability to automate the design of neural architectures tailored to specific tasks (Pham et al., 2018; Qin et al., 2021a; Wang et al., 2025). In particular, graph neural architecture search (GNAS) methods address the distinct challenge of modeling the intricate relationships between architectures and complex graph structures (Gao et al., 2021; Qin et al., 2022b; Guan et al., 2022; Zhang et al., 2023e; Xie et al., 2023). These methods can be broadly classified into three categories: reinforcement-learning-based approaches (Zhou et al., 2022b; Gao et al., 2022; 2023); evolutionary-based strategies (Nunes & Pappa, 2020; Li & King, 2020; Shi et al., 2022; Zhang et al., 2022a;b); and differentiable methods (Ding et al., 2021; Zheng et al., 2023; Huan et al., 2021; Zhang et al., 2023b;d; Qin et al., 2023; Yao et al., 2024; Ge et al., 2025), which enable continuous optimization of architectures within a differentiable search space. However, existing GNAS methods are limited in their ability to search for architectures for GNN-based GFMs.

### 6.3. Graph Invariant Representation Learning

Graph invariant presentation learning has emerged as a powerful approach for graph representation learning, focusing on capturing the stable relationships between graph data and tasks. Recent works have explored various applications of graph invariant learning in out-of-distribution generalization (Ma et al., 2019; Wu et al., 2022b; Li et al., 2022b;c; Zhang et al., 2022c; 2023c; 2024b; Li et al., 2024a). For instance, DIR (Wu et al., 2022b) discovers causal rationales that remain invariant across different distributions while filtering out spurious patterns that are unstable. DIDA (Zhang

et al., 2022c) leverages invariant structures and features with stable predictive performance across distribution shifts. However, these methods focus on capturing stable relationships for accurate label prediction. We apply this concept to architecture search, aiming to define invariant patterns that support stable architecture prediction, and design our method based on this concept.

### 6.4. Graph Self-supervised Learning

Graph self-supervised learning (SSL) has attracted significant attention in recent years, with numerous methods proposed to learn effective representations from graph data without relying on labeled information. These methods can be broadly classified into two categories: contrastive learning and generative learning. Contrastive learning approaches (You et al., 2020; Hassani & Khasahmadi, 2020; Li et al., 2021; Zhang et al., 2024a; Li et al., 2022d) aim to maximize the agreement between positive pairs of graph samples while minimizing it between negative pairs. In contrast, generative learning approaches (Tan et al., 2023; Xia et al., 2023; Hou et al., 2023) learn representations by reconstructing graph structures or attributes from partially observed data. These SSL techniques have demonstrated strong performance across a variety of graph-related tasks, including node classification, link prediction, and graph classification. In our work, we adopt a contrastive learning strategy to extract distinct patterns from diverse datasets, enabling the model to capture richer information and better identify architecture-specific requirements.

## 7. Conclusion

Existing graph neural architecture search methods fail to search for architectures for GNN-based GFMs. In this paper, we analyze the problem of *architecture inconsistency*, demonstrate that existing GNAS methods cannot effectively search for architectures for GNN-based GFMs, and tackle it by discovering an invariant graph-architecture relationship. We propose a novel Automated Graph Foundation Model with Adaptive Architecture Customization (**AutoGFM**) to search for graph neural architectures for each graph data with diverse tasks and domains individually. We introduce a disentangled contrastive graph encoder to discover invariant and variant patterns from graph data and an invariant-guided architecture customization module to customize graph neural architectures based on the discovered patterns. We also propose a curriculum architecture customization mechanism to mitigate the phenomenon of some particular data dominating the search process. Experimental results demonstrate that **AutoGFM** outperforms existing methods. One limitation of our work is that we mainly focus on graph neural architecture search on the GNN-based GFMs, and we leave the exploration of other types of GFMs for future work.

## Acknowledgements

This work is supported by National Natural Science Foundation of China No.62222209, Beijing National Research Center for Information Science and Technology under Grant No.BNR2023TD03006.

## Impact Statement

This paper presents work whose goal is to advance the field of Machine Learning. There are many potential societal consequences of our work, none which we feel must be specifically highlighted here.

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

# A. Proof

## A.1. Proof of Proposition 3.2

Proposition 3.2: If there exist two datasets $\mathcal{D}_i, \mathcal{D}_j \in \mathcal{D}$, the optimal operation for $\mathcal{D}_i$ is different from $\mathcal{D}_j$, the operations will render optimization conflicts.

*Proof.* Assume the optimal operation for $\mathcal{D}_i$ is $o_1$ and the optimal operation for $\mathcal{D}_j$ is $o_2$. The overall architecture function is defined as: $F(G) = o_1 f_1(G) + o_2 f_2(G)$. The MSE loss function $\mathcal{L}_{all}(F)$ is given by:

$$\mathcal{L}_{all}(F) = \mathcal{L}_i(F) + \mathcal{L}_j(F) + \mathcal{L}_k(F), \quad k \neq i, j, \tag{20}$$

where $\mathcal{L}_i(F) = \sum_k (F(G_{i,k}) - y_{i,k})^2$, $\mathcal{L}_j(F) = \sum_k (F(G_{j,k}) - y_{j,k})^2$.

We examine the effects of changes in $o_1$ and $o_2$ on the two terms $\mathcal{L}_i(F)$ and $\mathcal{L}_j(F)$ by calculating the partial derivatives of the two terms with respect to $o_1$ and $o_2$: $\frac{\partial \mathcal{L}_i}{\partial o_1}, \frac{\partial \mathcal{L}_i}{\partial o_2}, \frac{\partial \mathcal{L}_j}{\partial o_1}$, and $\frac{\partial \mathcal{L}_j}{\partial o_2}$.

We first simplify $\mathcal{L}_i(F)$ as follows:

$$\mathcal{L}_i(F) = \sum_k (F(G_{i,k}) - y_{i,k})^2 \tag{21}$$

$$= \sum_k (o_1 f_1(G_{i,k}) + o_2 f_2(G_{i,k}) - y_{i,k})^2 \tag{22}$$

$$= \sum_k (o_1 f_1(G_{i,k}) + o_2 f_2(G_{i,k}) - f_1(G_{i,k}))^2 \tag{23}$$

$$= \sum_k ((1 - o_2) f_1(G_{i,k}) + o_2 f_2(G_{i,k}) - f_1(G_{i,k}))^2 \tag{24}$$

$$= o_2^2 \sum_k (f_2(G_{i,k}) - f_1(G_{i,k}))^2 \tag{25}$$

$$= (1 - o_1)^2 \sum_k (f_2(G_{i,k}) - f_1(G_{i,k}))^2 . \tag{26}$$

Equation (22) expands $F(G_{i,k})$ as $F(G_{i,k}) = o_1 f_1(G_{i,k}) + o_2 f_2(G_{i,k})$. In Equation (23), $y_{i,k}$ is replaced with $f_1(G_{i,k})$ because the optimal operation for $\mathcal{D}_i$ is $o_1$, i.e., $y_{i,k} = f_1(G_{i,k})$. In Equation (24), $o_1$ is replaced with $1 - o_2$ due to the constraint $o_1 + o_2 = 1$. Equation (25) simplifies the equation.

Similarly, for $\mathcal{L}_j(F)$:

$$\mathcal{L}_j(F) = o_1^2 \sum_k (f_2(G_{j,k}) - f_1(G_{j,k}))^2$$

$$= (1 - o_2)^2 \sum_k (f_2(G_{j,k}) - f_1(G_{j,k}))^2 . \tag{27}$$

Then we examine the effects of changes in $o_1$ on the two terms $\mathcal{L}_i(F)$ and $\mathcal{L}_j(F)$ by calculating the partial derivatives of the two terms with respect to $o_1$: $\frac{\partial \mathcal{L}_i}{\partial o_1}, \frac{\partial \mathcal{L}_j}{\partial o_1}$.

$$\mathcal{L}_i = (1 - o_1)^2 \sum_k (f_2(G_{i,k}) - f_1(G_{i,k}))^2 . \tag{28}$$

$$\mathcal{L}_j = o_1^2 \sum_k (f_2(G_{j,k}) - f_1(G_{j,k}))^2 . \tag{29}$$

We calculate $\frac{\partial \mathcal{L}_i}{\partial o_1}, \frac{\partial \mathcal{L}_j}{\partial o_1}$ as follows:

$$\frac{\partial \mathcal{L}_i}{\partial o_1} = (2o_1 - 2) \sum_k \left(f_2(G_{i,k}) - f_1(G_{i,k})\right)^2 < 0 \tag{30}$$

$$\frac{\partial \mathcal{L}_j}{\partial o_1} = 2o_1 \sum_k \left(f_2(G_{j,k}) - f_1(G_{j,k})\right)^2 > 0 \tag{31}$$

$o_1, o_2 \in (0,1)$, and $f_1 \neq f_2$ so that $\sum_k \left(f_2(G_{i,k}) - f_1(G_{i,k})\right)^2$ and $\sum_k \left(f_2(G_{j,k}) - f_1(G_{j,k})\right)^2$ are both positive. Therefore, $\frac{\partial \mathcal{L}_i}{\partial o_1} < 0$ and $\frac{\partial \mathcal{L}_j}{\partial o_1} > 0$. To minimize the loss function $\mathcal{L}_i$, $o_1$ must be increased, whereas minimizing the loss function $\mathcal{L}_j$ requires decreasing $o_1$. Similar results about $o_2$ can be obtained for $\frac{\partial \mathcal{L}_i}{\partial o_2}, \frac{\partial \mathcal{L}_j}{\partial o_2}$. Therefore, the operation optimization objective for $\mathcal{D}_i$ is different from $\mathcal{D}_j$, *i.e.*, the operations will render optimization conflicts.

### A.2. Proof of Proposition 4.1

Proposition 4.1: if $P(A \mid Z_I, Z_V) = P(A \mid Z_I)$, the conditional mutual information $I(A, Z_V \mid Z_I)$ achieves its minimum value of 0: $I(A, Z_V \mid Z_I) = 0$

*Proof.* The conditional mutual information $I(A, Z_V \mid Z_I)$ is defined as:

$$I(A, Z_V \mid Z_I) = \mathbb{E}_{Z_I} \left[ \mathbb{E}_{A, Z_V \mid Z_I} \left[ \log \frac{P(A, Z_V \mid Z_I)}{P(A \mid Z_I)P(Z_V \mid Z_I)} \right] \right]. \tag{32}$$

Based on $P(A \mid Z_I, Z_V) = P(A \mid Z_I)$, we can obtain $P(A, Z_V \mid Z_I) = P(A \mid Z_I)P(Z_V \mid Z_I)$ as follows:

$$P(A \mid Z_I, Z_V) = P(A \mid Z_I), \tag{33}$$
$$P(A \mid Z_I, Z_V)P(Z_V \mid Z_I) = P(A \mid Z_I)P(Z_V \mid Z_I), \tag{34}$$
$$P(A, Z_V \mid Z_I) = P(A \mid Z_I)P(Z_V \mid Z_I). \tag{35}$$

Substituting the conditional independence into the definition of conditional mutual information, we obtain:

$$\begin{aligned} I(A, Z_V \mid Z_I) &= \mathbb{E}_{Z_I} \left[ \mathbb{E}_{A, Z_V \mid Z_I} \left[ \log \frac{P(A \mid Z_I)P(Z_V \mid Z_I)}{P(A \mid Z_I)P(Z_V \mid Z_I)} \right] \right] \\ &= \mathbb{E}_{Z_I} \left[ \mathbb{E}_{A, Z_V \mid Z_I} \left[ \log 1 \right] \right] \\ &= 0. \end{aligned} \tag{36}$$

Therefore, when $P(A \mid Z_I, Z_V) = P(A \mid Z_I)$, the conditional mutual information $I(A, Z_V \mid Z_I)$ achieves its minimum value of 0. This implies that, given $Z_I$, there is no additional dependence between A and $Z_V$.

## B. More Experiments

### B.1. Architecture Visualizations

To clearly visualize the customized architectures tailored to different datasets, we presented a heatmap in Figure 4, illustrating the choice weights of each operation at each layer. Firstly, we observe that different graph datasets prefer distinct architectures; for example, Cora mainly prefers GraphConv and GraphSAGE, whereas these two operations are rarely selected for PubMed. This observation further supports our earlier assumption that different datasets require different architectures, and some datasets exhibit inconsistent architectural preferences. Moreover, we find that many datasets prefer varying operations across different layers. For instance, the Arxiv dataset prefers GCN in the first layer and GAT in the second layer. Such fine-grained architectural preferences are challenging to meet through manual design, highlighting the advantage of automated, customized architectures.

### B.2. Hyperparameters Analysis

We analyze the sensitivity of the important hyperparameters $\lambda$ and $\beta$ in our method on the WikiCS dataset. We adjust $\lambda, \beta \in \{1e-1, 1e-2, 1e-3, 1e-4\}$, while maintaining the default value of the other hyperparameters unchanged.

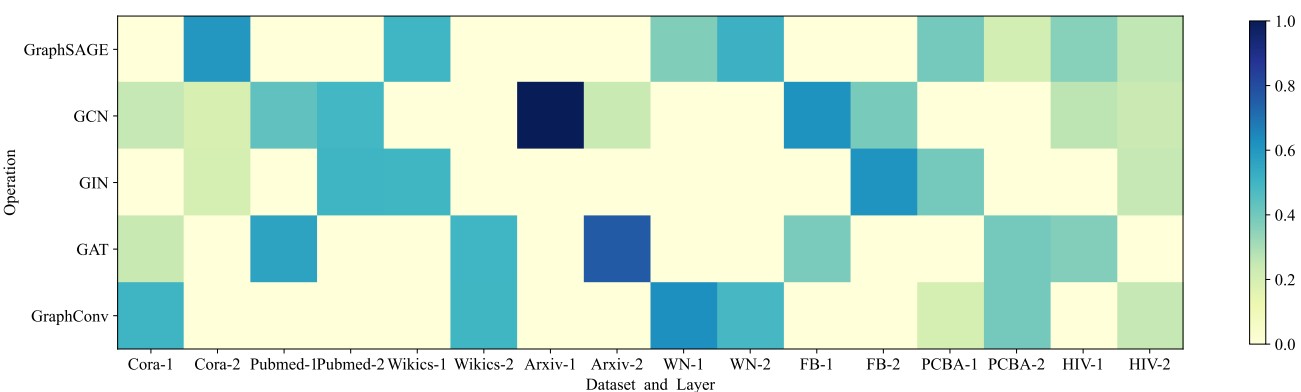

*Figure 4.* A showcase heatmap displays customized architectures for different datasets, where the number following the dataset name represents the architecture's layer.

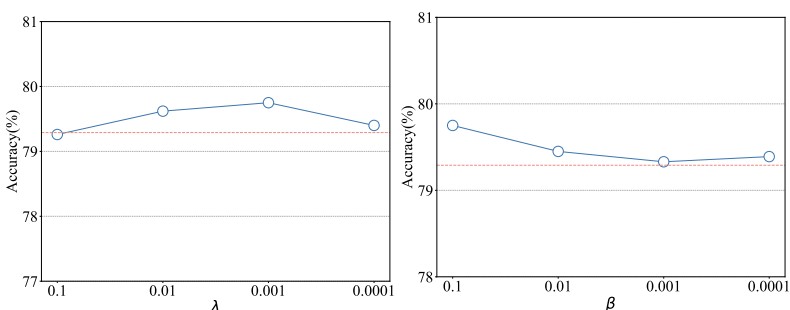

*Figure 5.* Hyperparameter sensitivity analysis on WikiCS dataset. The blue lines denote the results of our method and the red dashed lines are the results of the best baseline.

The results are shown in Figure 5. The hyperparameter $\lambda$ in Equation (19) controls the trade-off between $L_{task}$ and $L_{dis}$. Specifically, $L_{task}$ aims to maximize the mutual information between the invariant pattern $Z_I$ and the architecture A, ensuring that $Z_I$ is sufficient to predict A. In contrast, $L_{dis}$ aims to minimize the mutual information between the invariant pattern $Z_I$ and the variant pattern $Z_V$, thereby enabling the extraction of two disjoint patterns from the data. We adjust its value within the set $\{1e-1, 1e-2, 1e-3, 1e-4\}$. As shown in Figure 5, when $\lambda$ is set too low, the model's performance deteriorates, confirming that proper disentanglement of $Z_I$ and $Z_V$ is essential for effective architecture prediction. Conversely, when $\lambda$ is set too high, performance also declines, indicating that while ensuring the separation between the two patterns, it is equally important that the $Z_I$ retains sufficient information to predict the architecture. Overall, $\lambda$ is an important hyperparameter for balancing the sufficiency and disentanglement. The hyperparameter $\beta$ in Equation (19) controls the trade-off between $L_{task}$ and $L_{inv}$. Specifically, $L_{inv}$ aims to shield architecture A from the influence of $Z_V$ given the invariant pattern $Z_I$. As demonstrated in Figure 5, setting $\beta$ too low results in degraded model performance, underscoring the importance of effectively shielding A from the influence of $Z_V$ given $Z_I$. Thus, $\beta$ is also a critical hyperparameter for balancing the sufficiency and invariance conditions of the patterns captured by the model.

### B.3. More Few-shot Learning results

We conduct more N-way K-shot experiments on the few-shot learning task. The results are shown in Table 3 and Table 4. Our method outperforms the baselines mostly across various N-way K-shot settings, further verifying the effectiveness of the customized architectures.

*Table 3.* Accuracy (%) with std of different methods on Cora under N-way K-shot settings. The highest result is **bold**.

| Method | 7-way | | | 5-way | | | 2-way | | |
|---|---|---|---|---|---|---|---|---|---|
| | 5-shot | 3-shot | 1-shot | 5-shot | 3-shot | 1-shot | 5-shot | 3-shot | 1-shot |
| OFA | $32.10_{\pm1.79}$ | $36.03_{\pm2.11}$ | $30.38_{\pm2.39}$ | $42.28_{\pm2.35}$ | $31.28_{\pm2.63}$ | $23.68_{\pm1.67}$ | $72.20_{\pm3.82}$ | $62.22_{\pm1.17}$ | $51.85_{\pm4.35}$ |
| GFT | $43.55_{\pm7.43}$ | $43.31_{\pm8.11}$ | $41.40_{\pm8.04}$ | $52.30_{\pm6.57}$ | $51.47_{\pm6.33}$ | $49.80_{\pm6.79}$ | $75.00_{\pm4.08}$ | $76.33_{\pm3.56}$ | $72.92_{\pm4.64}$ |
| GCN | $42.57_{\pm7.50}$ | $42.38_{\pm7.42}$ | $43.07_{\pm7.37}$ | $45.00_{\pm7.40}$ | $44.53_{\pm7.93}$ | $44.80_{\pm8.94}$ | $71.58_{\pm5.20}$ | $71.42_{\pm4.03}$ | $70.58_{\pm4.01}$ |
| GAT | $47.71_{\pm8.02}$ | $47.31_{\pm7.78}$ | $46.12_{\pm7.10}$ | $52.30_{\pm6.05}$ | $51.73_{\pm7.32}$ | $50.17_{\pm7.41}$ | $75.92_{\pm3.89}$ | $75.17_{\pm5.36}$ | $72.83_{\pm5.48}$ |
| GraphSAGE | $42.40_{\pm6.12}$ | $42.07_{\pm6.12}$ | $40.50_{\pm6.11}$ | $51.17_{\pm5.13}$ | $50.80_{\pm5.34}$ | $49.50_{\pm5.55}$ | $74.20_{\pm2.95}$ | $74.33_{\pm2.47}$ | $72.08_{\pm5.89}$ |
| GIN | $47.24_{\pm7.33}$ | $47.02_{\pm7.32}$ | $45.29_{\pm6.26}$ | $49.83_{\pm7.79}$ | $49.17_{\pm8.10}$ | $48.97_{\pm6.73}$ | $75.25_{\pm8.60}$ | $\mathbf{76.83_{\pm8.36}}$ | $71.50_{\pm7.44}$ |
| GraphConv | $41.60_{\pm9.20}$ | $40.93_{\pm8.80}$ | $38.67_{\pm8.50}$ | $46.13_{\pm9.73}$ | $44.90_{\pm10.41}$ | $42.57_{\pm8.63}$ | $67.75_{\pm9.84}$ | $67.00_{\pm11.48}$ | $63.42_{\pm8.37}$ |
| Darts | $42.81_{\pm7.92}$ | $42.10_{\pm7.45}$ | $43.29_{\pm7.65}$ | $49.23_{\pm7.69}$ | $49.50_{\pm7.18}$ | $46.97_{\pm6.00}$ | $71.67_{\pm4.35}$ | $71.42_{\pm5.54}$ | $69.58_{\pm6.01}$ |
| GraphNAS | $41.62_{\pm9.20}$ | $40.60_{\pm9.23}$ | $38.64_{\pm8.31}$ | $50.07_{\pm6.66}$ | $50.13_{\pm7.29}$ | $47.30_{\pm5.92}$ | $71.83_{\pm4.21}$ | $73.17_{\pm5.33}$ | $68.00_{\pm5.69}$ |
| GASSO | $42.95_{\pm5.75}$ | $42.07_{\pm5.85}$ | $40.31_{\pm6.10}$ | $49.53_{\pm6.60}$ | $50.87_{\pm6.10}$ | $47.53_{\pm7.40}$ | $71.08_{\pm6.42}$ | $70.92_{\pm5.26}$ | $68.00_{\pm4.41}$ |
| GRACES | $47.57_{\pm7.22}$ | $46.31_{\pm7.42}$ | $45.43_{\pm6.73}$ | $50.17_{\pm7.74}$ | $49.30_{\pm6.12}$ | $49.40_{\pm6.20}$ | $74.81_{\pm5.82}$ | $74.42_{\pm5.47}$ | $72.58_{\pm4.90}$ |
| **Ours** | $\mathbf{47.76_{\pm8.06}}$ | $\mathbf{47.33_{\pm7.80}}$ | $\mathbf{46.29_{\pm7.24}}$ | $\mathbf{53.93_{\pm6.95}}$ | $\mathbf{52.50_{\pm6.84}}$ | $\mathbf{50.87_{\pm5.55}}$ | $\mathbf{76.43_{\pm5.45}}$ | $76.55_{\pm4.48}$ | $\mathbf{73.92_{\pm6.64}}$ |

*Table 4.* Accuracy (%) with std of different methods on WN18RR under N-way K-shot settings. The highest result is **bold**.

| Method | 10-way | | | 5-way | | | 3-way | | |
|---|---|---|---|---|---|---|---|---|---|
| | 5-shot | 3-shot | 1-shot | 5-shot | 3-shot | 1-shot | 5-shot | 3-shot | 1-shot |
| OFA | $32.64_{\pm1.56}$ | $30.56_{\pm1.02}$ | $25.82_{\pm1.07}$ | $48.32_{\pm3.19}$ | $45.04_{\pm2.39}$ | $34.40_{\pm1.47}$ | $60.72_{\pm3.82}$ | $61.29_{\pm2.56}$ | $51.77_{\pm2.65}$ |
| GFT | $35.50_{\pm4.59}$ | $35.50_{\pm5.02}$ | $35.33_{\pm4.20}$ | $48.80_{\pm3.61}$ | $48.53_{\pm3.68}$ | $48.13_{\pm4.37}$ | $62.56_{\pm2.71}$ | $60.67_{\pm3.93}$ | $58.44_{\pm3.84}$ |
| GCN | $30.40_{\pm3.02}$ | $29.78_{\pm3.64}$ | $29.85_{\pm4.14}$ | $44.70_{\pm2.99}$ | $44.97_{\pm3.95}$ | $44.77_{\pm3.45}$ | $54.06_{\pm5.36}$ | $53.33_{\pm5.64}$ | $53.28_{\pm4.77}$ |
| GAT | $34.70_{\pm3.15}$ | $34.37_{\pm3.43}$ | $34.50_{\pm2.98}$ | $46.23_{\pm4.44}$ | $46.33_{\pm4.50}$ | $46.30_{\pm4.43}$ | $59.56_{\pm3.85}$ | $59.39_{\pm3.45}$ | $58.06_{\pm4.34}$ |
| GraphSAGE | $38.30_{\pm2.16}$ | $38.17_{\pm2.34}$ | $38.03_{\pm2.03}$ | $48.10_{\pm3.78}$ | $47.83_{\pm3.88}$ | $47.90_{\pm3.66}$ | $62.39_{\pm3.48}$ | $61.44_{\pm3.48}$ | $59.39_{\pm3.50}$ |
| GIN | $37.47_{\pm3.10}$ | $36.92_{\pm4.03}$ | $36.62_{\pm4.17}$ | $47.57_{\pm5.56}$ | $47.80_{\pm5.29}$ | $47.60_{\pm3.81}$ | $61.33_{\pm5.98}$ | $61.83_{\pm6.35}$ | $58.22_{\pm4.93}$ |
| GraphConv | $39.62_{\pm3.44}$ | $39.28_{\pm2.27}$ | $38.93_{\pm3.77}$ | $48.40_{\pm3.38}$ | $47.63_{\pm2.51}$ | $46.07_{\pm3.23}$ | $61.39_{\pm4.11}$ | $60.06_{\pm3.60}$ | $59.33_{\pm3.43}$ |
| Darts | $38.65_{\pm3.55}$ | $38.57_{\pm3.22}$ | $37.22_{\pm2.68}$ | $47.40_{\pm4.36}$ | $46.03_{\pm3.61}$ | $46.43_{\pm3.20}$ | $60.17_{\pm2.32}$ | $58.78_{\pm4.63}$ | $57.89_{\pm3.62}$ |
| GraphNAS | $36.98_{\pm5.23}$ | $37.03_{\pm3.32}$ | $36.79_{\pm3.17}$ | $46.07_{\pm3.64}$ | $47.07_{\pm4.21}$ | $45.87_{\pm3.87}$ | $58.56_{\pm5.40}$ | $60.50_{\pm3.89}$ | $57.11_{\pm1.36}$ |
| GASSO | $37.37_{\pm3.31}$ | $37.42_{\pm1.91}$ | $37.13_{\pm3.52}$ | $47.90_{\pm4.14}$ | $47.77_{\pm3.63}$ | $46.20_{\pm3.24}$ | $59.61_{\pm4.94}$ | $59.83_{\pm5.20}$ | $57.83_{\pm4.06}$ |
| GRACES | $39.13_{\pm2.29}$ | $38.24_{\pm3.60}$ | $38.76_{\pm2.63}$ | $48.37_{\pm3.76}$ | $47.67_{\pm4.04}$ | $47.00_{\pm3.15}$ | $61.50_{\pm3.31}$ | $60.00_{\pm4.01}$ | $59.17_{\pm6.35}$ |
| **Ours** | $\mathbf{40.02_{\pm2.26}}$ | $\mathbf{39.55_{\pm2.46}}$ | $\mathbf{39.34_{\pm3.03}}$ | $\mathbf{49.93_{\pm3.63}}$ | $\mathbf{49.10_{\pm3.31}}$ | $\mathbf{48.47_{\pm4.38}}$ | $\mathbf{63.11_{\pm5.80}}$ | $\mathbf{61.94_{\pm2.61}}$ | $\mathbf{59.72_{\pm4.26}}$ |

## C. More Related Works

### C.1. LLM-based Graph Foundation Models.

LLM-based Graph Foundation Models employ LLMs as predictors within a unified generative framework for graph tasks (Liu et al., 2023c; Pan et al., 2024a; Fang et al., 2024; Jin et al., 2023; Zhu et al., 2024a; Yu et al.; Huang et al., 2023; 2024b). For instance, InstructGLM (Ye et al., 2023) employs a generative framework in which LLMs predict node labels by generating them based on the nodes' textual attributes. GraphGPT (Tang et al., 2024b) adapts LLMs for downstream graph tasks through instruction tuning, integrating natural language with a graph-text aligner to capture and convey structural graph information. These approaches present a promising direction for the development of GFMs, as LLMs can seamlessly unify the output of various graph tasks. Unlike GNNs, which require task-specific adjustments for model training, LLMs can accept a wide range of queries and generate appropriate responses. However, a key challenge lies in effectively translating graph structures into a format that LLMs can interpret. Current research tackles this problem with two primary approaches. The first involves describing the graph structure using natural language (graph to text) (Zhao et al., 2023; Fatemi et al., 2023; Zhao et al., 2023; Wang et al., 2024a). The second approach draws inspiration from Visual Language Models (VLMs), where the graph is first processed into embeddings using GNNs or projectors (graph to token), and an LLM then decodes the graph embeddings (Chen et al., 2024; Tian et al., 2024; Tang et al., 2024a). These methods demonstrate competence in fundamental reasoning tasks such as connectivity checks and cycle detection, but struggle with more complex graph tasks

that require capturing intricate graph patterns, such as graph classification.

## C.2. Graph Curriculum Learning

Curriculum learning is a training strategy that involves presenting training data in a meaningful order, typically starting with simpler examples and gradually progressing to more complex ones. This approach has been shown to improve the performance of various machine learning models (Bengio et al., 2009; Gong et al., 2015; Li et al., 2023; Chen et al., 2021;?; Zhang et al., 2022d; Zhou et al., 2022c;d; Chen et al., 2023; Zhou et al., 2023; Huang et al., 2024a; Zhou et al., 2024). Graph curriculum learning (GCL) is different from traditional curriculum learning due to the inherent dependencies of graph data(Gong et al., 2019; Zhou et al., 2022a; Wang et al., 2023; Wu et al., 2024). Researchers leverage graph structures to measure difficulty through predefined or automated strategies. For instance, CLNode(Wei et al., 2023) is a Curriculum Graph Learning method that measures local difficulty by considering the class diversity among a node's neighbors and uses global features to identify mislabeled nodes. RCL(Zhang et al., 2023a) gradually integrates node relationships into the training process, based on the complexity of those relationships. We utilize the concept of curriculum learning to enhance the architecture search process, mitigating the dominance of specific data on the search process.

## D. Experimental Setup

### D.1. Dataset

Table 5. Dataset statistics (Liu et al., 2023a).

| Dataset | Domain | Task | # Graphs | Avg. #Nodes | Avg. #Edges | # Classes |
|---------|--------|------|----------|-------------|-------------|-----------|
| **Cora** | Citation | Node | 1 | 2,708 | 10,556 | 7 |
| **PubMed** | Citation | Node | 1 | 19,717 | 44,338 | 3 |
| **Arxiv** | Citation | Node | 1 | 169,343 | 1,166,243 | 40 |
| **WikiCS** | Web link | Node | 1 | 11,701 | 216,123 | 10 |
| **FB15K237** | Knowledge | Link | 1 | 14,541 | 310,116 | 237 |
| **WN18RR** | Knowledge | Link | 1 | 40,943 | 93,003 | 11 |
| **PCBA** | Molecule | Graph | 437,929 | 26.0 | 28.1 | 128 |
| **HIV** | Molecule | Graph | 41,127 | 25.5 | 27.5 | 2 |
| **ChEMBL** | Molecule | Graph | 365,065 | 25.9 | 55.9 | 1,048 |

**Dataset Statistics.** We follow the preprocessing method described in (Liu et al., 2023a; Wang et al., 2024b), employing the Sentence Transformer (Reimers & Gurevych, 2019) to convert raw textual descriptions of nodes and edges into 768-dimensional features. For knowledge graphs (KGs), we do not transform edge textual information into edge features, as the existing textual information already provides sufficient knowledge for KG completion (Wang et al., 2024b). The statistics of the datasets are detailed in Table 5.

**Dataset Splitting.** We adopt the same splitting strategy as (Liu et al., 2023a; Wang et al., 2024b). For **Cora** and **PubMed** select 20 labeled nodes per class for training. We utilize a predefined set of 10 splits with different random seeds to compute the average performance. For **WikiCS**, we report the average accuracy over 20 distinct training splits, each generated with 20 different random seeds. In each split, 5% of the nodes from each class are used for training. For **Arxiv, HIV**, and **PCBA**, we employ the official dataset splits and conduct experiments 10 times using different random seeds to determine the average accuracy. The **FB15K237** dataset consists of 272,115 edges in the training set, 17,535 edges in the validation set, and 20,466 edges in the test set. Meanwhile, for **WN18RR**, the corresponding numbers are 86,835, 3,034, and 3,134, respectively. Each experiment is repeated 10 times with different random seeds, and the final results are reported as the average accuracy.

### D.2. Baseline

We compare our proposed **AutoGFM** with the following baselines and provide a brief description of each method:

- **Vanilla GNNs**
  - **GCN** (Kipf & Welling, 2017): Graph Convolutional Networks (GCN) utilizes graph convolutional layers to learn node representations by aggregating information from neighboring nodes.

- **GAT** (Velickovic et al., 2017): Graph Attention Networks (GAT) uses attention mechanisms to weigh the importance of neighboring nodes when updating node representations.
- **GIN** (Xu et al., 2018): Graph Isomorphism Network (GIN) employs a sum aggregation function and a learnable MLP to update node representations, aiming to achieve maximum discriminative power among graph structures.

- **Self-supervised GNNs**

    - **BGRL** (Thakoor et al., 2021): BGRL leverages a bootstrap-style contrastive learning approach without negative samples, maximizing agreement between online and target networks over augmented graph views.
    - **GraphMAE** (Hou et al., 2022): GraphMAE employs masked autoencoders to learn node representations by reconstructing masked node features from the input graph.
    - **GIANT** (Chien et al., 2022): GIANT is a self-supervised GNN framework that performs multi-granularity contrastive learning across multiple graphs to learn generalizable node representations.

- **GFMs**

    - **OFA** (Liu et al., 2023a): OFA is a GNN-based foundation model that integrates LLMs to process textual features across different domains and unifies graph-related tasks through subgraph classification.
    - **GFT** (Wang et al., 2024b): GFT is a GNN-based foundation model that incorporates computation trees to identify transferable patterns across graph structures, enabling the learning of generalizable representations for various graph domains and tasks.

- **Manually designed GNNs**

    - **GraphSAGE** (Hamilton et al., 2017): GraphSAGE learns node representations in an inductive manner by sampling and aggregating features from a node's local neighborhood using various aggregation functions such as mean or LSTM.
    - **GraphConv** (Morris et al., 2019): GraphConv is a generalized graph convolutional operator that combines node features with their neighbors' features using a trainable transformation, capturing local structure in the graph.

- **GNAS methods**

    - **DARTS** (Liu et al., 2018): DARTS is a differentiable architecture search framework that relaxes the search space into a continuous domain, enabling efficient gradient-based optimization of neural architectures.
    - **GraphNAS** (Gao et al., 2021): GraphNAS applies reinforcement learning to search for optimal GNN architectures by modeling the architecture design process as a sequential decision-making problem.
    - **GASSO** (Qin et al., 2021b): GASSO enables differentiable architecture search via gradient descent and discovers more effective graph neural architectures by incorporating graph structure learning as a denoising process during the search procedure.
    - **Graces** (Qin et al., 2022a): Graces achieves generalization under distribution shifts by designing instance-specific GNN architectures tailored to the unknown distribution of each graph.

For **Vanilla GNNs**, **self-supervised methods**, and **GFMs**, we reproduce the results based on their original papers and publicly available code. To ensure a fair comparison between **manually designed GNNs** and **GNAS** baselines, we employ GFT (Wang et al., 2024b) as the base model. Specifically, for **manually designed GNNs**, we replace the GNN in GFT with various manually designed GNNs and follow identical pretraining and finetuning procedures as GFT. For **GNAS** methods, we substitute the GNN component in GFT with different GNAS methods. Architecture search is performed during the pretraining stage, whereas in the finetuning stage, we further optimize only the parameters of the searched architectures without additional architecture searches.

### D.3. Hyperparameters

We evaluate different GNN architectures and GNAS methods based on GFT (Wang et al., 2024b), following the default hyperparameters of GFT to maintain consistency. To ensure a fair comparison, we set the dimensionality of all methods to 768, use the same search space and operations (GCN, GIN, GAT, GraphSAGE, GraphConv), and fix the number of layers to 2. For our method, we explore hyperparameter $\lambda, \beta \in \{1e-1, 1e-2, 1e-3, 1e-4\}$ and empirically select $\lambda$ and $\beta$. The

learning rate of the disentangled contrastive graph encoder is set to $5e - 3$, and the learning rate of the architecture predictor is set to $3e - 2$. The dimensionality of both the graph encoder and the supernet is 768. Each experiment is conducted 10 times, and we report the average performance along with standard deviations.

### D.4. Configurations

We conduct all experiments on the following configurations:

- **Operating System**: Ubuntu 20.04.5 LTS

- **CPU**: Intel(R) Xeon(R) Gold 5218R CPU @ 2.10GHz

- **GPU**: NVIDIA A100-SXM4-40GB and NVIDIA A100-SXM4-80GB

- **Software**: Python 3.9, CUDA 12.2, PyTorch (Paszke et al., 2019) 1.13.1.

