# OpenReview forum: "AutoGFM: Automated Graph Foundation Model with Adaptive Architecture Customization"
_ICML.cc/2025/Conference — ICML 2025 oral_

### Official Review · Reviewer_YeCs · 2025-03-12

**Overall Recommendation:** 4

**Summary:**

This paper introduces an automated graph foundation model with adaptive graph neural architecture customization. The authors address the architecture inconsistency problem in graph foundation models. The proposed method consists of graph encoder, architecture customization, and curriculum training. The theoretical analysis and empirical results on multiple datasets seem to show the effectiveness of the proposed approach.

**Claims And Evidence:**

The claims regarding architecture inconsistency and the benefits of the proposed method are grounded with theoretical justifications and experimental comparisons. The authors demonstrate that fixed architectures underperform on diverse datasets and propose a method to adapt architectures dynamically.

**Essential References Not Discussed:**

The paper could benefit from citing more recent advances in self-supervised learning techniques for GNN adaptation.

**Experimental Designs Or Analyses:**

The experimental setup is comprehensive, covering multiple datasets and comparisons. The results indicate superior performance.

**Methods And Evaluation Criteria:**

The proposed methodology is sound, leveraging contrastive learning and mutual information constraints. The evaluation benchmarks are relevant, with a reasonable selection of baselines. However, the choice of hyperparameters for different datasets is not well discussed, and sensitivity analysis should be enhanced.

**Other Comments Or Suggestions:**

Clarify the hyperparameter tuning process and discuss more on ablation studies.

**Other Strengths And Weaknesses:**

A key strength is the novel combination of graph foundation models and GraphNAS, which opens a new direction on this topic.

**Questions For Authors:**

How does the proposed method compare in terms of computational efficiency with other NAS-based approaches?

**Relation To Broader Scientific Literature:**

The paper builds on prior work in graph foundation models and GraphNAS, providing a novel adaptation mechanism. The references are mostly relevant.

**Theoretical Claims:**

The theoretical claims, particularly the propositions regarding architecture inconsistency, are mathematically sound.

---

> ### Author Rebuttal · Authors · 2025-03-31
>
> We would like to express our sincere gratitude to the reviewer for the detailed comments and insightful questions. We respond to each of the reviewer’s comments point by point as follows.
>
> > 1. "Clarify the hyperparameter tuning process and discuss more on ablation studies."
>
> Thank you for bringing this to our attention. Regarding the hyperparameter tuning process, we pretrain the GFA model with $\lambda,\beta \in \\{1e-1,1e-2,1e-3,1e-4\\}$. Subsequently, we fine-tune these pretrained models and evaluated their performance on the validation set to empirically determine the hyperparameters.
>
> Additionally, we provide a further analysis of the ablation studies below.
>
> (i) The disentangled contrastive graph encoder module is designed to extract discriminative invariant and variant patterns from the data by pulling similar samples closer and pushing dissimilar samples apart in the latent space. Removing this module impairs the extraction of invariant patterns and reduces the distinguishability between patterns extracted from different datasets, ultimately harming the effectiveness of architecture prediction.
>
> (ii) The invariant-guided architecture customization module serves to shield architecture $A$ from the influence of variant patterns $Z_V$ given the invariant pattern $Z_I$. The substantial performance decrease observed upon removing this module highlights the importance of effectively isolating architecture predictions from $Z_V$ influences, reinforcing the critical role of this module in ensuring the invariance conditions of captured patterns.
>
> (iii) This curriculum architecture customization mechanism aims to reduce data dominance in the architecture search process. Removing this module causes certain operations, which perform well on specific datasets during early training stages, to dominate the search process. Consequently, other datasets may neglect potentially beneficial operations.
>
>
> > 2. "How does the proposed method compare in terms of computational efficiency with other NAS-based approaches?"
>
> We are grateful for your feedback. In the original manuscript, we analyze the time complexity of GFA as ${O}(|E|d_e +|V|d_e^2 +|\\mathcal{O}|^2d_e +|\\mathcal{O}|(|E|d_a +|V|d_a^2))$. Considering the term with the largest complexity, this is approximately ${O}(|\\mathcal{O}|(|E|d +|V|d^2))$. The time complexity for most existing GNN methods is typically ${O}(|E|d +|V|d^2)$, and GNAS methods also exhibit a approximate complexity of ${O}(|\\mathcal{O}|(|E|d +|V|d^2))$. Thus, our method's complexity is comparable to existing GNAS approaches.
>
>
> > 3. "However, the choice of hyperparameters for different datasets is not well discussed, and sensitivity analysis should be enhanced."
>
> Thank you for your suggestion. Since GFA is trained jointly across all datasets, the hyperparameters $\lambda$ and $\beta$ are consistent for all datasets. Specifically, we set $\lambda$ to $1e-3$ and $\beta$ to $1e-1$.
>
> We provide a more detailed discussion on the hyperparameter sensitivity. The hyperparameter $\lambda$ in Eq.17 controls the trade-off between $L_{task}$ and $L_{dis}$. Specifically, $L_{task}$ aims to maximize the mutual information between the invariant pattern $Z_I$ and the architecture $A$, ensuring that $Z_I$ is sufficient to predict $A$. In contrast, $L_{dis}$ aims to minimize the mutual information between the invariant pattern $Z_I$ and the variant pattern $Z_V$, thereby enabling the extraction of two disjoint patterns from the data. We adjust its value within the set $\\{1e-1,1e-2,1e-3,1e-4\\}$. As shown in Figure 5, when $\lambda$ is set too low, the model's performance deteriorates, confirming that proper disentanglement of $Z_I$ and $Z_V$ is essential for effective architecture prediction. Conversely, when $\lambda$ is set too high, performance also declines, indicating that while ensuring the separation between the two patterns, it is equally important that the $Z_I$ retains sufficient information to predict the architecture. Overall, $\lambda$ is an important hyperparameter for balancing the sufficiency and disentanglement. The hyperparameter $\beta$ in Eq.17 controls the trade-off between $L_{task}$ and $L_{inv}$. Specifically, $L_{inv}$ aims to shield architecture $A$ from the influence of $Z_V$ given the invariant pattern $Z_I$. As demonstrated in Figure 5, setting $\beta$ too low results in degraded model performance, underscoring the importance of effectively shielding $A$ from the influence of $Z_V$ given $Z_I$. Thus, $\beta$ is also a critical hyperparameter for balancing the sufficiency and invariance conditions of the patterns captured by the model.
>
> > 4. "more recent advances in self-supervised learning techniques for GNN adaptation."
>
> We sincerely appreciate your valuable suggestion. In response, we will include a paragraph in the Related Works section of our revised manuscript, discussing recent advances in self-supervised learning techniques for GNN adaptation.

---

> > ### Comment · Reviewer_YeCs · 2025-04-06
> >
> > Thanks to the authors for the rebuttal. The authors’ responses have addressed my concerns. I would like to raise my overall assessment to this work.

---

### Official Review · Reviewer_Qn8E · 2025-03-13

**Overall Recommendation:** 5

**Summary:**

The paper introduces a framework for adapting GNN architectures dynamically to improve generalization in GFMs. Existing graph neural architecture search methods struggle to design architectures for GNN-based GFMs. This paper addresses the issue of architecture inconsistency by identifying an invariant relationship between graphs and architectures. The authors propose GFA, an automated approach that tailors GNN architectures to different graph datasets, tasks, and domains.

The key contributions of the paper include:

- Proposed a disentangled contrastive graph encoder to extract invariant and variant patterns from graph data.
- Proposed an invariant-guided architecture customization strategy to tailor GNN architectures in a dynamic way.
- Proposed a curriculum-based architecture customization mechanism to mitigate the effects of data domination during the search process.
- Provided theoretical insights demonstrating the limitations of existing GNAS methods in handling architecture inconsistency.
- Conducted extensive experiments on multiple datasets showing that GFA outperforms baseline methods.

**Claims And Evidence:**

The primary claim that adaptive architectures enhance performance across various graph settings is supported by theoretical analysis and empirical results. The paper provides mathematical proofs demonstrating why existing GNAS methods struggle with architecture inconsistency, emphasizing the need for dynamic customization. Additionally, experimental evaluations on diverse datasets show significant improvements over both manually designed GNNs and existing GNAS approaches. Ablation studies seem to validate the effectiveness of the proposed method. But the analyses on the results are a little limited.

**Essential References Not Discussed:**

I did not locate essential references missing.

**Experimental Designs Or Analyses:**

The evaluation is thorough, but the study would benefit from detailed ablation experiments to analyze the impact of various components of the proposed method.

**Methods And Evaluation Criteria:**

The methodology is clearly presented and builds upon solid foundations in NAS and GNN customization. The experimental setup/evaluation is comprehensive. Table 1 covered all common graph tasks, like node classification, link prediction and graph classification.

**Other Comments Or Suggestions:**

It would be useful to expand the discussion on Figure 4 and explain what key insights can be drawn regarding how different architectures adapt to various datasets.

**Other Strengths And Weaknesses:**

Strengths:

- Strong empirical validation with extensive experiments across multiple datasets.
- Novel use of curriculum learning to mitigate data domination effects in GNAS.
- Theoretical insights into architecture inconsistency and its impact on GNAS.
- Comprehensive experimental design, including ablation studies and few-shot learning evaluation.

Weaknesses:

- The discussions on the results shown in the figures or tables could be enhanced.
- The framework figure 2 is a little simple and did not clearly show the technical details of the proposed method.
- The descriptions on the inference stage are missing.

**Questions For Authors:**

1. Can you explain what claims you want to support with the showcase Figure 4? What important points can be derived from the showcase? I am not very clear to that.

2. How does the computational complexity of GFA compare to classical GNNs and standard GNAS methods?

3. Could you show the pipeline of GFA duration the inference stage? The current algorithm and method description only focus on the training stage.

**Relation To Broader Scientific Literature:**

The work follows the recent trends in adaptive neural architecture design and graph foundation model.

**Theoretical Claims:**

The theoretical contributions are valuable. The proofs are correct after my careful check. The authors provide mathematical formulations and proofs that highlight the architecture inconsistency problem in existing GNAS methods. Their theoretical analysis shows that under the assumption that different datasets require distinct architectures, differentiable GNAS methods (e.g., DARTS) fail due to optimization conflicts. My verification confirms the correctness of these proofs, and the proposed invariant-guided architecture customization is a theoretically sound solution to this problem.

---

> ### Author Rebuttal · Authors · 2025-03-31
>
> We would like to express our sincere appreciation to the reviewer for providing us with detailed suggestions. We have carefully reviewed each comment and offer the following responses.
>
> > 1. "Can you explain what claims you want to support with the showcase Figure 4?"
>
> Thank you for highlighting this point. To clearly visualize the customized architectures tailored to different datasets, we presented a heatmap in Figure 4, illustrating the choice weights of each operation at each layer. Firstly, we observe that different graph datasets prefer distinct architectures; for example, Cora mainly prefers GraphConv and GraphSAGE, whereas these two operations are rarely selected for PubMed. This observation further supports our earlier assumption that different datasets require different architectures, and some datasets exhibit inconsistent architectural preferences. Moreover, we find that many datasets prefer varying operations across different layers. For instance, the Arxiv dataset prefers GCN in the first layer and GAT in the second layer. Such fine-grained architectural preferences are challenging to meet through manual design, highlighting the advantage of automated, customized architectures.
>
> > 2. "How does the computational complexity of GFA compare to classical GNNs and standard GNAS methods?"
>
> We are grateful for your feedback. In the original manuscript, we analyzed the complexity of GFA as ${O}(|E|d_e +|V|d_e^2 +|\\mathcal{O}|^2d_e +|\\mathcal{O}|(|E|d_a +|V|d_a^2))$. Considering the term with the largest complexity, this is approximately ${O}(|\\mathcal{O}|(|E|d +|V|d^2))$. The complexity for most existing GNN methods is typically ${O}(|E|d +|V|d^2)$, and GNAS methods also exhibit a approximate complexity of ${O}(|\\mathcal{O}|(|E|d +|V|d^2))$. Thus, our method's complexity is comparable to existing GNAS approaches.
>
> > 3. "Could you show the pipeline of GFA duration the inference stage?"
>
> Thank you for highlighting this point. During the inference stage, given an input graph, we first utilize the Disentangled Contrastive Graph Encoder to obtain its invariant pattern representation, denoted as $Z_I$. Then, $Z_I$ is fed into the Invariant Predictor within the Invariant Guided Architecture Customization module to generate a customized architecture. This customized architecture is subsequently employed as the GNN component within the GFM to perform prediction.
>
> > 4. "Ablation studies seem to validate the effectiveness of the proposed method. But the analyses on the results are a little limited."
>
> Thank you for bringing this to our attention. We provide a further analysis of the ablation studies below.
>
> (i) The disentangled contrastive graph encoder module is designed to extract discriminative invariant and variant patterns from the data by pulling similar samples closer and pushing dissimilar samples apart in the latent space. Removing this module impairs the extraction of invariant patterns and reduces the distinguishability between patterns extracted from different datasets, ultimately harming the effectiveness of architecture prediction.
>
> (ii) The invariant-guided architecture customization module serves to shield architecture $A$ from the influence of variant patterns $Z_V$ given the invariant pattern $Z_I$. The substantial performance decrease observed upon removing this module highlights the importance of effectively isolating architecture predictions from $Z_V$ influences, reinforcing the critical role of this module in ensuring the invariance conditions of captured patterns.
>
> (iii) This curriculum architecture customization mechanism aims to reduce data dominance in the architecture search process. Removing this module causes certain operations, which perform well on specific datasets during early training stages, to dominate the search process. Consequently, other datasets may neglect potentially beneficial operations.
>
> > 5. "The discussions on the results shown in the figures or tables could be enhanced."
>
> Thank you for your suggestion. We will add more discussions on the results shown in the figures and tables in our revised manuscript to make our paper more readable.
>
> > 6. "The framework figure 2 is a little simple and did not clearly show the technical details of the proposed method."
>
> Thank you for highlighting this point. We have refined our framework by incorporating additional technical details to better align with the content and equations presented in the Method section, specifically in the following three aspects.
>
> (1) Enhancing the clarity of the pipeline by adding more numerical indexing and clear directional arrows to improve readability.
>
> (2) Including additional equations directly in the figures, enabling readers to easily identify corresponding equations from the text.
>
> (3) Reflecting the unique advantages of our method, such as the customization of different architectures tailored to specific datasets, and the progressive architecture search process driven by curriculum learning.

---

### Official Review · Reviewer_Dusn · 2025-03-13

**Overall Recommendation:** 4

**Summary:**

The authors introduce GFA, a framework for graph neural network architecture customization in graph foundation models. The paper addresses the architecture inconsistency problem, which arises when different graph domains and tasks require varying GNN architectures. To tackle this, the authors propose a disentangled contrastive graph encoder, an invariant-guided architecture customization strategy, and a curriculum-based optimization mechanism to improve architecture search for diverse datasets. They conduct extensive experiments across real-world datasets, showing that GFA outperforms state-of-the-art baselines. Furthermore, the paper provides theoretical analysis demonstrating the limitations of existing graph neural architecture search methods in handling architecture inconsistency. In summary, the paper presents a novel approach to automated GNN-based GFMs with architecture search.

## update after rebuttal

I will keep my positive opinion towards the paper after rebuttal.

**Claims And Evidence:**

The main claim of the paper is that customizing GNN architectures according to different datasets improves GFM performance. This claim is supported by empirical evidence, including evaluations on eight diverse datasets. The paper effectively shows that fixed architectures used in prior GFMs lead to suboptimal performance, while GFA dynamically adapts architectures, achieving state-of-the-art results. A key contribution is the theoretical analysis that demonstrates why standard differentiable GNAS methods struggle under architecture inconsistency. This analysis, backed by proofs in Appendix, provides strong theoretical motivation for the proposed approach.
However, while the theoretical insights are technically solid, they could be presented more intuitively to help reader understanding.

**Essential References Not Discussed:**

No more works should be referenced.

**Experimental Designs Or Analyses:**

The experimental results are strong, showing improvements over baselines. However, the few-shot experiments in table 2 seem to be confusing (I will explain it in weaknesses). Also, the paper does not discuss how hyperparameter sensitivity affects performance thoroughly.

**Methods And Evaluation Criteria:**

The methodology is convincing. The three core modules, disentangled contrastive graph encoder, invariant-guided customization, and curriculum-based optimization, are integrated to address architecture inconsistency issue. The experimental design is comprehensive, covering datasets across node, edge, and graph classification tasks. The authors compare against multiple baselines, including vanilla GNNs, self-supervised learning methods, existing GFMs, and various GNAS techniques.

**Other Comments Or Suggestions:**

-	The experiment should consider additional way settings beyond the current specific configurations.
-	The authors should provide more detailed descriptions of the baseline methods.
-	The framework diagram (Figure 2) should be revised to clarify the technical details of the proposed GFA.

**Other Strengths And Weaknesses:**

I think the paper did a good job at these points:

-	It proposes a novel, end-to-end approach for GNN architecture customization in GFMs.
-	It shows theoretical analysis exposing limitations of existing GNAS methods.
-	It integrates theoretical, empirical, and algorithmic innovations into a unified framework.

But I still have some concerns on the paper:

-	One concern relates to the “N-way K-shot” few-shot learning experiments. The authors explore different way settings (Cora-7 way, WN18RR-10 way, CHEMHIV-2 way), but it is unclear why these specific values of N were chosen for each dataset. A clearer justification would be beneficial.
-	Another concern is on the hyperparameter sensitivity. The paper does not discuss how hyperparameter sensitivity affects performance thoroughly.
-	And there are limited descriptions of the baseline methods.

**Questions For Authors:**

How about the performance in additional way settings as in table 2? Does the method still perform best in these new few-shot settings?

**Relation To Broader Scientific Literature:**

The work aligns with ongoing research in graph LLMs and graph foundation models.

**Theoretical Claims:**

The theoretical aspect of this paper addresses why standard differentiable GNAS methods struggle to manage architectural inconsistencies across heterogeneous graph datasets. The authors provide a series of theorems (with proofs in Appendix) showing how prior methods might overlook essential dataset-specific structures when optimizing a universal, shared architecture parameter space. The derivations appear mathematically solid. Nonetheless, additional intuitive explanations could make the theoretical arguments more readable for a broader audience.

---

> ### Author Rebuttal · Authors · 2025-03-31
>
> We sincerely appreciate the insightful comments provided by the reviewer. We have carefully considered each point raised and would like to respond as follows.
>
> > 1. "The experiment should consider additional way settings beyond the current specific configurations."
>
> Thank you for raising this important point. We have conducted additional experiments to validate our model’s performance under more N-way settings on the Cora and WN18RR datasets. Due to character limits, we only include a subset of the most competitive baselines below. The complete results will be provided in our second-round response and included in the revised manuscript.
>
> |Cora||5 way|||2-way||
> |-|-|-|-|-|-|-|
> ||5-shot|3-shot|1-shot|5-shot|3-shot|1-shot|
> |GAT|52.30±6.05|51.73±7.32|50.17±7.41|75.92±3.89|75.17±5.36|72.83±5.48|
> |GIN|49.83±7.79|49.17±8.10|48.97±6.73|75.25±8.60|**76.83±8.36**|71.50±7.44|
> |GRACES|50.17±7.74|49.30±6.12|49.40±6.20|74.81±5.82|74.42±5.47|72.58±4.90|
> |Ours|**53.93±6.95**|**52.50±6.84**|**50.87±5.55**|**76.43±5.45**|76.55±4.48|**73.92±6.64**|
>
>
> |WN18RR||5 way|||3-way||
> |-|-|-|-|-|-|-|
> ||5-shot|3-shot|1-shot|5-shot|3-shot|1-shot|
> |GAT|46.23±4.44|46.33±4.50|46.30±4.43|59.56±3.85|59.39±3.45|58.06±4.34|
> |GIN|47.57±5.56|47.80±5.29|47.60±3.81|61.33±5.98|61.83±6.35|58.22±4.93|
> |GRACES|48.37±3.76|47.67±4.04|47.00±3.15|61.50±3.31|60.00±4.01|59.17±6.35|
> |Ours|**49.93±3.63**|**49.10±3.31**|**48.47±4.38**|**63.11±5.80**|**61.94±2.61**|**59.72±4.26**|
>
> Our method outperforms the baselines mostly across various N-way K-shot settings, further verifying the effectiveness of the customized architectures.
>
> > 2. "additional intuitive explanations could make the theoretical arguments more readable for a broader audience."
>
> Thank you for your suggestion. We provide some additional intuitive explanations below.
>
> **Assumption 3.1** assumes that the optimal architectures required by two different datasets may differ. As shown in Figure 1(the performance of each architecture on different datasets), GCN achieves optimal performance on the PubMed dataset, while GraphSAGE performs best on the Wikics dataset.
>
> **Assumption 3.1** serves as a prerequisite condition for **Proposition 3.2**.
> **Proposition 3.2** demonstrates that when two datasets require different optimal architectures, current mainstream GNAS methods encounter optimization conflicts for GFM. As previously illustrated, the optimal architectures for PubMed and Wikics differ. Consequently, when existing GNAS methods search simultaneously for an architecture optimal for both datasets, they fail to identify a single architecture that performs best for both and are forced to compromise.
>
> **Assumption 3.3** defines what constitutes an invariant pattern for architecture prediction.
> - **Condition 1** indicates that the data contains two types of patterns: an invariant pattern $Z_I$, which reliably predicts the architecture, and a variant pattern $Z_V$, which cannot stably predict the architecture.
> - **Condition 2** highlights that the variant pattern $Z_V$ are not independent of the architecture $A$.
> - **Condition 3** states that, given the invariant pattern $Z_I$, the architecture $A$ is independent of the variant pattern $Z_V$, and $Z_I$ is sufficient for predicting $A$.
>
> **Proposition 4.1** aims to demonstrate that our method satisfies condition 3 of an invariant pattern for architecture prediction via enforcing  $P(A \\mid Z_{I}, Z_{V}\) = P(A \\mid Z_{I})$.
>
> > 3. "the paper does not discuss how hyperparameter sensitivity affects performance thoroughly."
>
> Thank you for bringing this to our attention. Due to character limitations, we will provide a more detailed discussion on hyperparameter sensitivity in our revised manuscript. Alternatively, we kindly refer the reviewer to our response to **Reviewer YeCs’s Question 3**, where we address a similar question. We apologize for any inconvenience this may cause.
>
> > 4. "The authors should provide more detailed descriptions of the baseline methods."
>
> We appreciate your suggestion. We will include a new section in the appendix providing detailed descriptions of all the baselines.
>
> > 5. "The framework diagram (Figure 2) should be revised to clarify the technical details of the proposed GFA."
>
> Thank you for highlighting this point. We have refined our framework by incorporating additional technical details to better align with the content and equations presented in the Method section, specifically in the following three aspects.
>
> (1) Enhancing the clarity of the pipeline by adding more numerical indexing and clear directional arrows to improve readability.
>
> (2) Including additional equations directly in the figures, enabling readers to easily identify corresponding equations from the text.
>
> (3) Reflecting the unique advantages of our method, such as the customization of different architectures tailored to specific datasets, and the progressive architecture search process driven by curriculum learning.

---

> > ### Comment · Reviewer_Dusn · 2025-04-04
> >
> > I have gone through all the reviews and rebuttals. I believe the paper is of high quality and have raised my score to 4.

---

### Official Review · Reviewer_QHCx · 2025-03-14

**Overall Recommendation:** 5

**Summary:**

This paper explores automated graph neural architecture search (GNAS) for Graph Foundation Models (GFMs) to overcome the limitations of fixed, hand-designed GNN architectures, which result in suboptimal performance across diverse graph domains and tasks. The authors identify the architecture inconsistency problem, where the optimal GNN architectures vary across different domains and tasks. To tackle this, they propose an Automated Graph Foundation Model with Adaptive Graph Neural Architecture Customization (GFA), which incorporates: a disentangled contrastive graph encoder to learn both invariant and variant patterns from graph data, an invariant-guided architecture customization strategy to adapt GNN architectures to different domains and tasks, and a curriculum architecture customization mechanism to mitigate the dominance of particular data during the search process. Additionally, the paper provides theoretical insights into the limitations of existing GNAS methods under the architecture inconsistency problem. Extensive experiments demonstrate that GFA outperforms baseline models, achieving state-of-the-art performance. This work is the first to address the problem of GNAS for GFMs.

**Claims And Evidence:**

The main point that fixed GNN architectures lead to suboptimal performance in diverse settings is justified through both theoretical analysis and empirical validation.

**Essential References Not Discussed:**

The discussion of related work is comprehensive. More recent advances in GFMs [1] could be relevant references.
[1] Graph Foundation Models: Concepts, Opportunities and Challenges. 	ArXiv:2310.11829.

**Experimental Designs Or Analyses:**

The experimental setup is comprehensive, covering multiple datasets and a set of baseline models, including manually designed GNNs, existing GNAS methods, and state-of-the-art Graph Foundation Models.

**Methods And Evaluation Criteria:**

The proposed method is evaluated on multiple datasets and makes sense.

**Other Comments Or Suggestions:**

As shown above, the authors should talk about the validity of the assumptions used in the method. Why do the authors need the Assumption 3.1 and Assumption 3.3? Are these assumptions common in the literature?
Minors: There are some typos in the manuscript. The corresponding symbols (periods or commas) after equations are missing in Eq. (4) (5) (14) (15) or wrong in Eq. (16).  Initial letters should be capitalized in line 434.

**Other Strengths And Weaknesses:**

Strengths:
-	The paper presents an important and novel problem—GNAS for Graph Foundation Models (GFMs)—which has not been previously explored. The identified architecture inconsistency problem is a significant contribution to the field.
-	The proposed GFA framework systematically addresses key challenges in GNAS for GFMs. The design of disentangled contrastive learning, invariant-guided customization, and curriculum-based customization is innovative.
-	The paper provides a theoretical analysis of the limitations of existing GNAS methods under the architecture inconsistency problem.
-	The empirical results demonstrate the effectiveness of GFA, showing state-of-the-art performance across multiple benchmarks.
Weaknesses:
-	The author did not explain whether the assumptions can be valid in the real world.
-	The writing can be improved. For example, more details in Sec. D.2 should be added for reproducing the results. The descriptions on the baselines are too simple, which brings difficulty to the readers that do not very familiar with this topic.

**Questions For Authors:**

1) Could you clarify whether the assumptions in the theories can be valid in the real world?
2) Could you add more details on the baselines? How do you implement the baselines? The current descriptions are too simple.

**Relation To Broader Scientific Literature:**

The work fits within GNAS and GNN literature.

**Theoretical Claims:**

The theoretical analysis is rigorous. The authors present a formulated argument demonstrating the optimization conflicts caused by architecture inconsistency in existing methods. The proof of Proposition 3.2 convincingly shows that a one-size-fits-all architecture search approach is insufficient for diverse graph tasks. Proposition 4.1 provides a justification for the proposed invariant-guided architecture customization strategy. But I would say that the theoretical analysis is based on some assumptions. It should be discussed whether these assumptions can be true in the real world.

---

> ### Author Rebuttal · Authors · 2025-03-31
>
> We would like to express our sincere gratitude to the reviewer for providing us with detailed comments and insightful questions. We have carefully considered the reviewer's feedback and would like to address each point as follows.
>
> > 1. "Could you clarify whether the assumptions in the theories can be valid in the real world?"
>
> Thank you for raising this important point. **Assumption 3.1** serves as a prerequisite condition for **Proposition 3.2**. Specifically, **Assumption 3.1** assumes that optimal architectures required by two different datasets may differ. To validate this assumption, we evaluated various GNN architectures based on a GNN-based GFM (GFT [1]) across multiple real-world datasets. Figure 1 presents a heatmap visualization of each architecture’s performance on different datasets, showing that optimal architectures indeed vary according to the dataset. For instance, GCN achieves optimal performance on the PubMed, while GraphSAGE performs best on the Wikics.
>
> **Assumption 3.3** defines what constitutes an invariant pattern for architecture prediction. The concept of the invariant pattern is well-defined, and methods based on this concept have been validated as effective in various real-world applications, such as academic citation networks [2] and molecular structures [3]. Unlike previous work, which has primarily focused on capturing stable relationships for accurate label prediction, we applied this concept to architecture search, aiming to define invariant patterns that support stable architecture prediction, and designed our method based on this concept.
>
> >  2. "Could you add more details on the baselines? How do you implement the baselines? The current descriptions are too simple."
>
> Thank you for bringing up this point. For **Vanilla GNNs**, **self-supervised methods**, and **GFMs**, we reproduce the results based on their original papers and publicly available code. To ensure a fair comparison between **manually designed GNNs** and **GNAS** baselines, we employ GFT[1] as the base model. Specifically, for **manually designed GNNs**, we replace the GNN in GFT with various manually designed GNNs and follow identical pretraining and finetuning procedures as GFT. For **GNAS** methods, we substitute the GNN component in GFT with different GNAS methods. Architecture search is performed during the pretraining stage, whereas in the finetuning stage, we further optimize only the parameters of the searched architectures without additional architecture searches. Furthermore, we use the same search space for both GNAS baselines and GFA, including operations such as GCN, GAT, GraphSAGE, GIN, and GraphConv within a super-network depth of 2 layers. We set the dimensionality of all methods to 768.
>
> >  3. "More recent advances in GFMs [1] could be relevant references. [1] Graph Foundation Models: Concepts, Opportunities and Challenges. ArXiv:2310.11829".
>
> Thank you for your suggestion. We will add this citation to the related work section of the revised manuscript.
>
> > 4. "The writing can be improved. For example, more details in Sec. D.2 should be added for reproducing the results. The descriptions on the baselines are too simple, which brings difficulty to the readers that do not very familiar with this topic."
>
> Thank you for your suggestion. We have expanded the description in Sec.D.2 as follows.
>
> We evaluate different GNN architectures and GNAS methods based on GFT[1], following the default hyperparameters of GFT to maintain consistency. To ensure a fair comparison, we set the dimensionality of all methods to 768, use the same search space and operations (GCN, GIN, GAT, GraphSAGE, GraphConv), and fix the number of layers to 2.  For our method, we explore hyperparameter $\lambda,\beta \in \\{1e-1,1e-2,1e-3,1e-4\\}$ and empirically set $\lambda$ to $1e-3$ and $\beta$ to $1e-1$. The learning rate of the disentangled contrastive graph encoder is set to $5e-3$, and the learning rate of the architecture predictor is set to $3e-2$. The dimensionality of both the graph encoder and the supernet is 768. Each experiment is conducted 10 times, and we report the average performance along with standard deviations.
>
> We will also include a new section in the appendix with detailed descriptions of all the baselines.
>
> > 5. "Minors: There are some typos in the manuscript. The corresponding symbols (periods or commas) after equations are missing in Eq. (4) (5) (14) (15) or wrong in Eq. (16). Initial letters should be capitalized in line 434."
>
> We sincerely appreciate your thoughtful observation. We have corrected these typos and will carefully revise the manuscript to address any remaining errors.
>
> [1] GFT: Graph Foundation Model with Transferable Tree Vocabulary
>
> [2] Learning Invariant Representations of Graph Neural Networks via Cluster Generalization
>
> [3] Learning Invariant Molecular Representation in Latent Discrete Space

---

> > ### Comment · Reviewer_QHCx · 2025-04-07
> >
> > Thanks for the response. My concerns have been addressed. This paper sounds good both in theory and in pracice. I'd like to raise the score.

---

### Decision · Program_Chairs · 2025-05-01

**Decision:**

Accept (oral)

**Comment:**

This paper proposes to use automated graph neural architecture search (GNAS) for Graph Foundation Models (GFMs) on diverse graph domains and tasks. To find an invariant graph-architecture relationship across domains and tasks, it presents an Automated Graph Foundation Model with Adaptive Graph Neural Architecture Customization (GFA). GFA consists of an encoder to distinguish invariant and variant patterns, an invariant-guided architecture customization strategy, and a curriculum architecture customization mechanism. Theoretical analysis and experiments illustrate the superiority of the proposed model. The claims are supported by clear and convincing evidence. The proposed method is novel and makes sense. This seems to be the first work to explore the problem of GNAS for GFMs. Thus, it is inspiring and possesses a high impact on the field. After the rebuttal, most concerns raised by reviewers were alleviated, and most reviewers raised their ratings.